# Signatures of Bayesian inference emerge from energy-efficient synapses

**James Malkin[1]\*, Cian O'Donnell[1,2†], Conor J Houghton[1†], Laurence Aitchison[1]**

[1]Faculty of Engineering, University of Bristol, Bristol, United Kingdom; [2]Intelligent Systems Research Centre, School of Computing, Engineering, and Intelligent Systems, Ulster University, Derry/Londonderry, United Kingdom

\*For correspondence:
james.malkin@bristol.ac.uk

†These authors contributed equally to this work

**Competing interest:** The authors declare that no competing interests exist.

**Abstract** Biological synaptic transmission is unreliable, and this unreliability likely degrades neural circuit performance. While there are biophysical mechanisms that can increase reliability, for instance by increasing vesicle release probability, these mechanisms cost energy. We examined four such mechanisms along with the associated scaling of the energetic costs. We then embedded these energetic costs for reliability in artificial neural networks (ANNs) with trainable stochastic synapses, and trained these networks on standard image classification tasks. The resulting networks revealed a tradeoff between circuit performance and the energetic cost of synaptic reliability. Additionally, the optimised networks exhibited two testable predictions consistent with pre-existing experimental data. Specifically, synapses with lower variability tended to have (1) higher input firing rates and (2) lower learning rates. Surprisingly, these predictions also arise when synapse statistics are inferred through Bayesian inference. Indeed, we were able to find a formal, theoretical link between the performance-reliability cost tradeoff and Bayesian inference. This connection suggests two incompatible possibilities: evolution may have chanced upon a scheme for implementing Bayesian inference by optimising energy efficiency, or alternatively, energy-efficient synapses may display signatures of Bayesian inference without actually using Bayes to reason about uncertainty.

## eLife assessment

This **important** study provides deep insight into a ubiquitous, but poorly understood, phenomenon: synaptic noise (primarily due to failures). Through a combination of theoretical analysis, simulations, and comparison to existing experimental data, this paper makes a **compelling** case that synapses are noisy because reducing noise is expensive. It touches on probably the most significant feature of living organisms -- their ability to learn -- and will be of broad interest to the neuroscience community.

## Introduction

The synapse is the major site of inter-cellular communication in the brain. The amplitude of synaptic postsynaptic potentials (PSPs) are usually highly variable or stochastic. This variability arises primarily presynaptically: the release of neurotransmitter from presynaptically housed vesicles into the synaptic cleft has variable release probabilities and variable quantal sizes (*Lisman and Harris, 1993*; *Branco and Staras, 2009*; *Brock et al., 2020*). Unreliable synaptic transmission seems puzzling, especially in light of evidence for low-noise, almost failure-free transmission at some synapses (*Paulsen and Heggelund, 1994*; *Paulsen and Heggelund, 1996*; *Bellingham et al., 1998*). Moreover, the degree to which a synapse is unreliable does not just vary from one synapse type to another, there is also an heterogeneity of precision amongst synapses of the same type (*Murthy et al., 1997*; *Dobrunz and*

*Stevens, 1997*). Given that there is capacity for more precise transmission, why is this capacity not used in more synapses?

Unreliable transmission degrades accuracy but *Laughlin et al., 1998*, showed that the synaptic connection from a photoreceptor to a retinal large monopolar cell could increase its precision by increasing the number of synapses, averaging the noise away, but this comes at the cost of extra energy per bit of information transmitted. Moreover, *Levy and Baxter, 2002*, demonstrated that there is a value for the precision which optimises the energy cost of information transmission. In this paper, we explore this notion of a performance-energy tradeoff.

However, it is important to consider precision and energy cost in the context of neuronal computation; the brain does not simply transfer information from neuron to neuron, it performs computation through the interaction between neurons. However, models outlining a synaptic energy-performance tradeoff, (*Laughlin et al., 1998*; *Levy and Baxter, 2002*; *Goldman, 2004*; *Harris et al., 2012*; *Harris et al., 2019*; *Karbowski, 2019*), predominantly consider information transmission between just two neurons and the corresponding information-theoretic view treats the synapse as an isolated conduit of information (*Shannon, 1948*). In contrast, in reality, a single synapse is just one unit of the computational machinery of the brain. As such, the performance of an individual synapse needs to be considered in the context of circuit performance. To perform computation in an energy-efficient way the circuit as a whole needs to allocate resources across different synapses to optimise the overall energy cost of computation (*Yu et al., 2016*; *Schug et al., 2021*).

Here, we consider the consequences of a tradeoff between network performance and energetic reliability costs that depend explicitly upon synapse precision. We estimate the energy costs associated with precision by considering the biological mechanisms underpinning synaptic transmission. By including these costs in a neural network designed to perform a classification task, we observe a heterogeneity in synaptic precision and find that this 'allocation' of precision is related to signatures of synapse 'importance', which can be understood formally on the grounds of Bayesian inference.

## Results

We proposed energetic costs for reliable synaptic transmission and then measured their consequences in an artificial neural network (ANN).

### Biophysical costs

Here, we seek to understand the biophysical energetic costs of synaptic transmission, and how those costs relate to the reliability of transmission (*Figure 1a*). We start by considering the underlying mechanisms of synaptic transmission. In particular, synaptic transmission begins with the arrival of a spike at the axon terminal. This triggers a large influx of calcium ions into the axon terminal. The increase in calcium concentration causes the release of neurotransmitter-filled vesicles docked at axonal release sites. The neurotransmitter diffuses across the synaptic cleft to the postsynaptic dendritic membrane. There, the neurotransmitter binds with ligand-gated ion channels causing a change in voltage, i.e., a PSP. This process is often quantified using the *Katz and Miledi, 1965*, quantal model of neurotransmitter release. Under this model, for each connection between two cells, there are $n$ docked, readily releasable vesicles (see *Figure 1a* for an illustration of a single synaptic connection with multi-vesicular release).

An alternative interpretation of this model might consider $n$ the number of uni-vesicular connections between two neurons. When the presynaptic cell spikes, each docked vesicle releases with probability $p$ and each released vesicle causes a PSP of size $q$. Thus, the mean, $\mu$, and variance, $\sigma^2$, of the PSP can be written (see *Figure 1b*),

$$\mu = npq$$
$$\sigma^2 = np(1-p)q^2 \tag{1}$$

where $q$ is considered a scaling variable. An assertion in our model is that variability in PSP strength is the result of variable numbers of vesicle release, not variability in $q$; here, during any PSP, $q$ is assumed constant across vesicles. While there is some suggestion that intra- and inter-site variability in $q$ is a significant component of PSP variability (see *Silver, 2003*), we ultimately expect quantal variability to be small relative to the variability attributed to vesicular release. This is supported by the classic

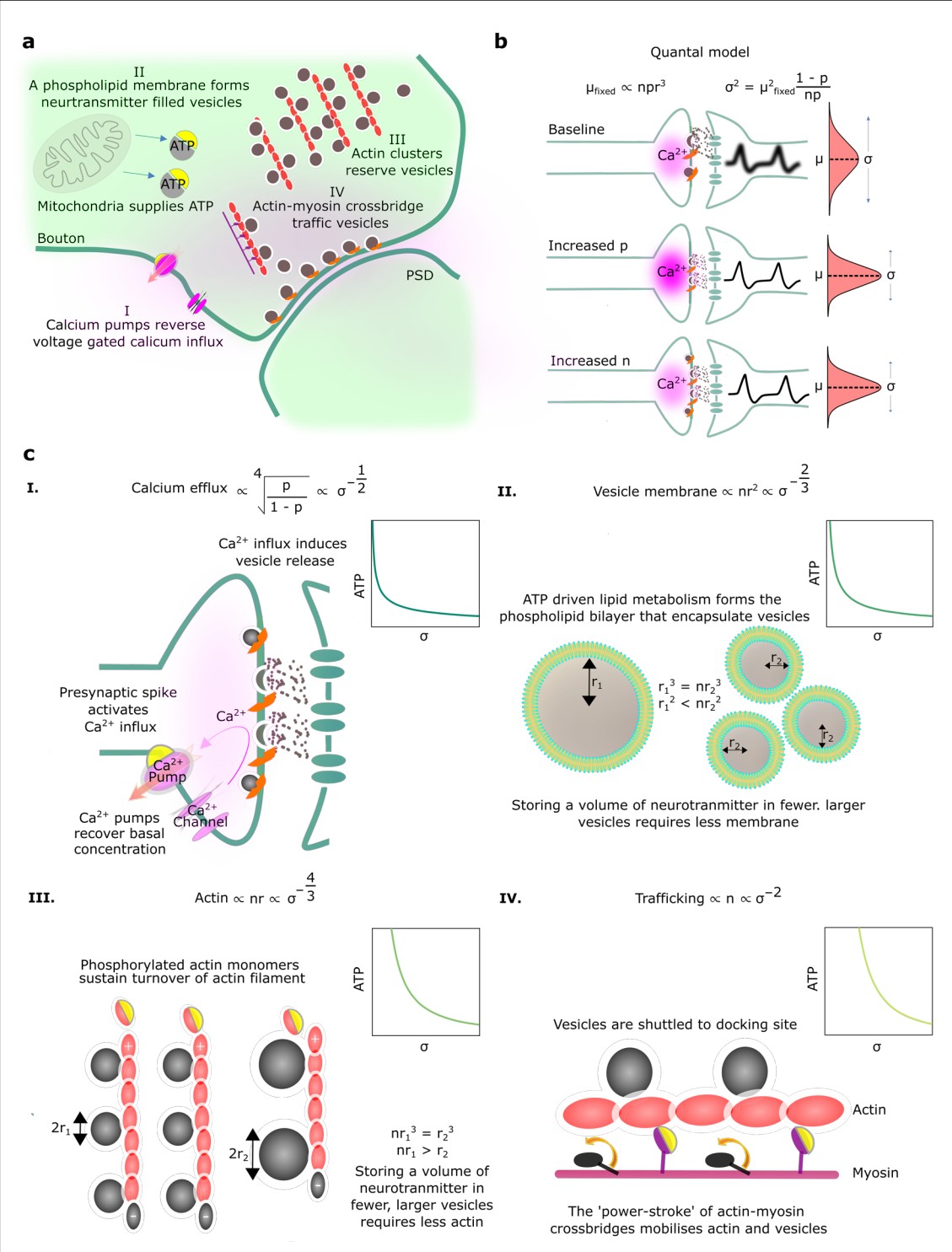

**Figure 1.** Physiological reliability costs. (**a**) Physiological processes that influence postsynaptic potential (PSP) precision. (**b**) A binomial model of vesicle release. For fixed PSP mean, increasing $p$ or $n$ decreases PSP variance. We have substituted $q \propto r^3$ to reflect that vesicle volume scales quantal size (**Karunanithi et al., 2002**). (**c**) Four different biophysical costs of reliable synaptic transmission. (I) Calcium pumps reverse the calcium influx that triggers vesicle release. A high probability of vesicle release requires a large influx of calcium, and extruding this calcium is costly. Note that $r$ represents the vesicle radius. (II) An equivalent volume of neurotransmitter can be stored in few large vesicles or shared between many smaller vesicles. Sharing a fixed volume of neurotransmitter among many small vesicles reduces PSP variability but increases vesicle surface area, creating greater demand for phospholipid metabolism and hence greater energetic costs. (III) Actin filament supports the structure of vesicle clusters at the terminal. Many and large vesicles require more actin and higher rates of ATP-dependent actin turnover. (IV) There are biophysical costs that scale with the number of vesicles (**Laughlin et al., 1998**; **Attwell and Laughlin, 2001**), e.g., vesicle trafficking driven by myosin-V active transport along actin filaments.

observation that PSP amplitude histograms have a multi-peak structure (*Boyd and Martin, 1956*; *Holler et al., 2021*), and by more direct measurement and modelling of vesicle release (*Forti et al., 1997*; *Raghavachari and Lisman, 2004*).

We considered four biophysical costs associated with improving the reliability of synaptic transmission, while keeping the mean fixed, and derived the associated scaling of the energetic cost with PSP variance.

*Calcium efflux*. Reliability is higher when the probability of vesicle release, $p$, is higher. As vesicle release is triggered by an increase in intracellular calcium, greater calcium concentration implies higher release probability. However, increased calcium concentration implies higher energetic costs. In particular, calcium that enters the synaptic bouton will subsequently need to be pumped out. We take the cost of pumping out calcium ions to be proportional to the calcium concentration, and take the relationship between release probability and calcium concentration to be governed by a Hill Equation, following *Sakaba and Neher, 2001*. The resulting relationship between energetic costs and reliability is $\text{cost} \propto \sigma^{-1/2}$ (*Figure 1cI*; see Appendix 1, 'Reliability costs' for further details).

*Vesicle membrane surface area*. There may also be energetic costs associated with producing and maintaining a large amount of vesicle membrane. *Purdon et al., 2002*, argues that phospholipid metabolism may take a considerable proportion of the brain's energy budget. Additionally, costs associated with membrane surface area may arise because of leakage of hydrogen ions across vesicles (*Pulido and Ryan, 2021*). Importantly, a cost for vesicle surface area is implicitly a cost on reliability. In particular, we could obtain highly reliable synaptic release by releasing many small vesicles, such that stochasticity in individual vesicle release events averages out. However, the resulting many small vesicles have a far larger surface area than a single large vesicle, with the same mean PSP. Thus, a cost on surface area implies a relationship between energetic costs and reliability; in particular $\text{cost} \propto \sigma^{-2/3}$ (*Figure 1cII*; see Appendix 1, 'Reliability costs' for further details).

*Actin*. Another cost for small but numerous vesicles arises from a demand for structural organisation of the vesicles pool by filaments such as actin (*Cingolani and Goda, 2008*; *Gentile et al., 2022*). Critically, there are physical limits to the number of vesicles that can be attached to an actin filament of a given length. In particular, if vesicles are smaller we can attach more vesicles to a given length of actin, but at the same time, the total vesicle volume (and hence the total quantity of neurotransmitter) will be smaller (*Figure 1cIII*). A fixed cost per unit length of actin thus implies a relationship between energetic costs and reliability of, $\text{cost} \propto \sigma^{-4/3}$ (see Appendix 1, 'Reliability costs').

*Trafficking*. A final class of costs is proportional to the number of vesicles (*Laughlin et al., 1998*). One potential biophysical mechanism by which such a cost might emerge is from active transport of vesicles along actin filaments or microtubles to release sites (*Chenouard et al., 2020*). In particular, vesicles are transported by ATP-dependent myosin-V motors (*Bridgman, 1999*), so more vesicles require a greater energetic cost for trafficking. Any such cost proportional to the number of vesicles gives rise to a relationship between energetic cost and PSP variance of the form, $\text{cost} \propto \sigma^{-2}$ (*Figure 1cIV*; see Appendix 1, 'Reliability costs').

*Costs related to PSP mean/magnitude*. While costs on precision are the central focus of this paper, it is certainly the case that other costs relating to the mean PSP magnitude constitute a major cost of synaptic transmission. For example, high amplitude PSPs require a large quantity of neurotransmitter, high probability of vesicle release, and a large number of postsynaptic receptors (*Attwell and Laughlin, 2001*). These can be formalised as costs on the PSP mean, $\mu$, and can additionally be related to L1 weight decay in a machine learning context (*Rosset and Zhu, 2006*; *Sacramento et al., 2015*).

## Reliability costs in ANNs

Next, we sought to understand how these biophysical energetic costs of reliability might give rise to patterns of variability in a trained neural network. Specifically, we trained ANNs using an objective that embodied a tradeoff between performance and reliability costs,

$$\text{Overall cost} = \text{performance cost} + \text{magnitude cost} + \text{reliability cost.} \qquad (2)$$

The 'performance cost' term measures the network's performance on the task, for instance in our classification tasks we used the usual cross-entropy cost. The 'magnitude cost' term captures costs

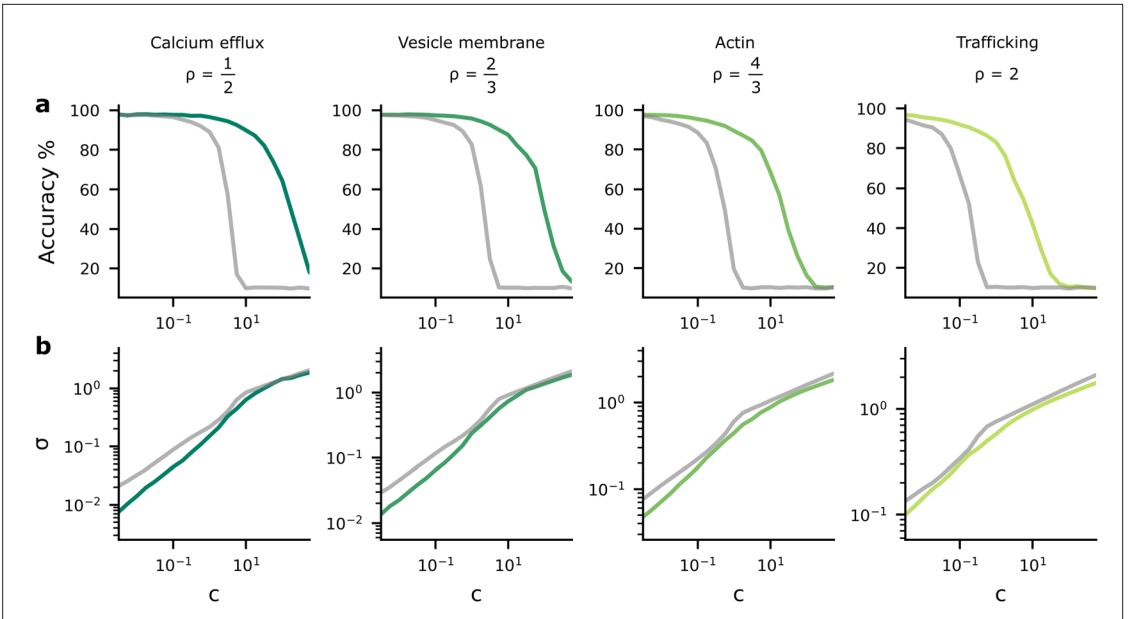

**Figure 2.** Accuracy and postsynaptic potential (PSP) variance as we change the tradeoff between reliability and performance costs. We changed the tradeoff by modifying $c$, in **Equation 4**, which multiplies the reliability cost. (**a**) As the reliability cost multiplier, $c$, increases, the accuracy decreases considerably. The green lines show the heterogeneous noise setting where the noise level is optimised on a per-synapse basis, while the grey lines show the homogeneous noise setting, where the noise is optimised, but forced to be the same for all synapses. (**b**) When the reliability cost multiplier, $c$, increases, the synaptic noise level (specifically, the average standard deviation, $\sigma$) increases.

that depend on the PSP mean, while the 'reliability cost' term captures costs that depend on the PSP precision. In particular,

$$\text{magnitude cost} = \lambda \sum_i |\mu_i|, \tag{3}$$

$$\text{reliability cost} = c \sum_i \sigma_i^{-\rho}. \tag{4}$$

Here, $i$ indexes synapses, and recall that $\sigma_i$ is the standard deviation of the $i$th synapse. The multiplier $c$ in the reliability cost determines the strength of the reliability cost relative to the performance cost. Small values for $c$ imply that the reliability cost term is less important, permitting precise transmission and higher performance. Large values for $c$ give greater importance to the reliability cost encouraging energy efficiency by allowing higher levels of synaptic noise, causing detriment to performance (see **Figure 2**).

We trained fully connected, rate-based neural network to classify MNIST digits. Stochastic synaptic PSPs were sampled from a Normal distribution,

$$w_i \sim \text{Normal}(\mu_i, \sigma_i). \tag{5}$$

where, recall, $\mu_i$ is the PSP mean and $\sigma_i^2$ is the PSP variance for the $i$th synapse. The output firing rate was given by

$$\text{firing rate} = f\left(\sum_i w_i x_i - w_0\right). \tag{6}$$

Here, $\sum_i w_i x_i - w_0$ can be understood as the somatic membrane potential, and $f$ represents the relationship between somatic membrane potential and firing rate; we used ReLU (**Fukushima, 1975**). We optimised network parameters $\mu_i$ and $\sigma_i$ using Adam (**Kingma and Ba, 2014**) (see Materials and methods for details on architecture and hyperparameters).

## The tradeoff between accuracy and reliability costs in trained networks

Next we sought to understand how the tradeoff between accuracy and reliability cost manifests in trained networks. Perhaps the critical parameter in the objective (*Equation 2* and *Equation 4*) was $c$, which controlled the importance of the reliability cost relative to the performance cost. We trained networks with a variety of different values of $c$, and with four values for $\rho$ motivated by the biophysical costs (the different columns).

In practice all the reliability costs and others we may have overlooked should together constitute an overall energetic reliability cost. However, it is difficult to estimate the specific contributions of different costs, i.e., the individual values of $c$. While *Attwell and Laughlin, 2001*; *Engl and Attwell, 2015*, estimate the ATP demands for various synaptic processes, it is difficult to relate these to the relative scale of each cost at a synapse level. Therefore, for simplicity, we kept each cost separate, training neural networks with just one choice of reliability cost; emphasising results shared across all costs. It is possible that one cost dominates all the others, but if that is not the case it will be necessary to use a more complicated reliability cost. However, since we have considered four costs with very different power-law behaviours, it is likely the behaviour will not be significantly different to what we have observed.

As expected, we found that as $c$ increased, performance fell (*Figure 2a*) and the average synaptic standard deviation increased (*Figure 2b*). Importantly, we considered two different settings. First, we considered an homogeneous noise setting, where $\sigma_i$ is optimised but kept the same across all synapses (grey lines). Second, we considered an heterogeneous noise setting, where $\sigma_i$ is allowed to vary across synapses, and is optimised on a per-synapse basis. We found that heterogeneous noise

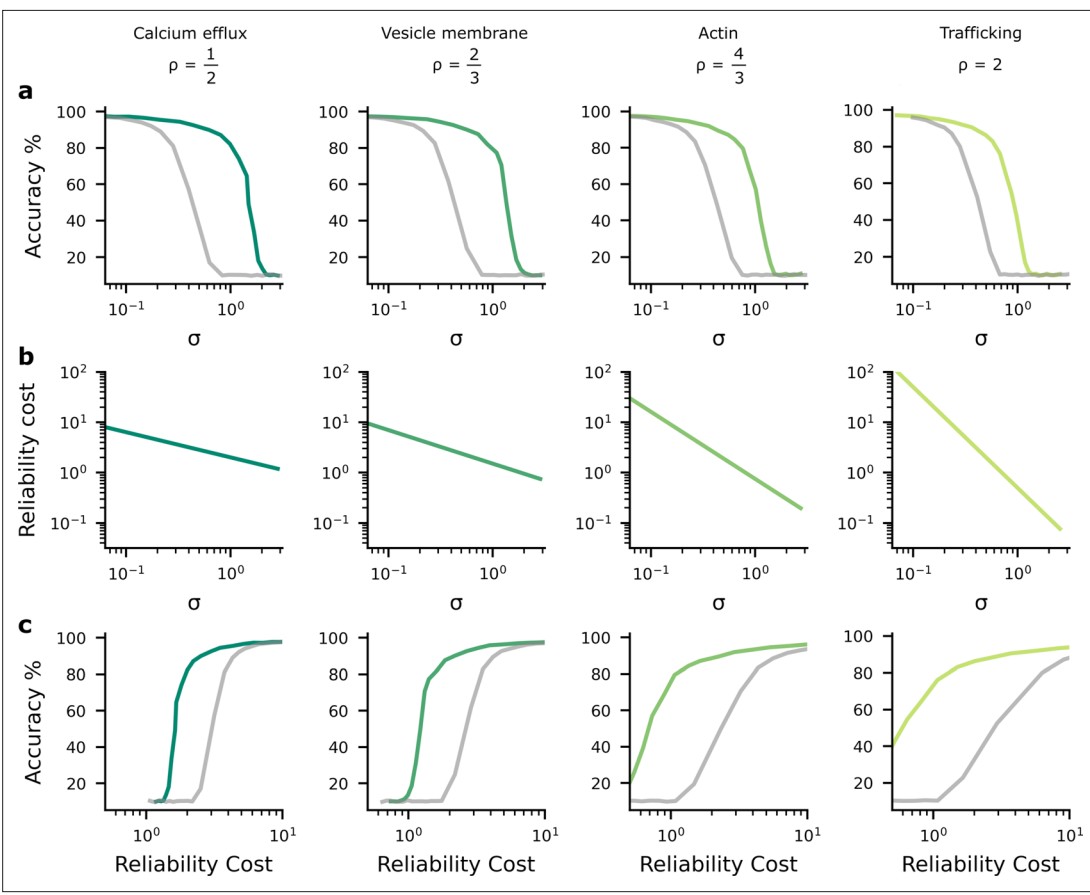

**Figure 3.** The performance-reliability cost tradeoff in artificial neural network (ANN) simulations. (**a**) Accuracy decreases as the average postsynaptic potential (PSP) standard deviation, $\sigma$, increases. The grey lines are for the homogeneous noise setting where the PSP variance is optimised but isotropic (i.e. the same across all synapses), while the green lines are for the heterogeneous noise setting, where the PSP variances are optimised individually on a per-synapse basis. (**b**) Increasing reliability by reducing $\sigma^2$ leads to greater reliability costs, and this relationship is different for different biophysical mechanisms and hence values for $\rho$ (columns). (**c**) Higher accuracy therefore implies larger reliability cost.

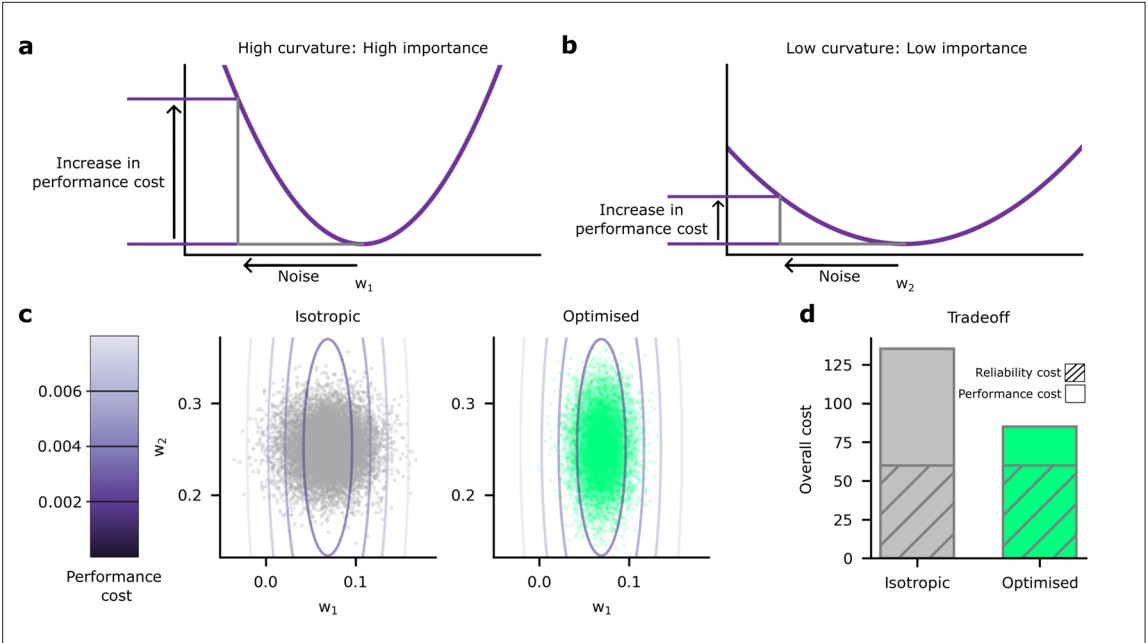

**Figure 4.** Schematic depiction of the impact of synaptic noise on synapses with different importance. (**a**) First, we considered an important synapse for which small deviations in the weight, $w_1$, e.g., driven by noise, imply a large increase in the performance cost. This can be understood as a high curvature of the performance cost as a function of $w_1$. (**b**) Next we considered an unimportant synapse, for which deviations in the weights cause far less increase in performance cost. (**c**) A comparison of the impacts of homogeneous and optimised heterogeneous variability for synapses $w_1$ and $w_2$ from (**a and b**). The performance cost is depicted using the purple contours, and realisations of the postsynaptic potentials (PSPs) driven by synaptic variability are depicted in the grey/green points. The grey points (left) depict homogeneous noise while the green points (right) depict optimised, heterogeneous noise. (**d**) The noise distributions in panel c are chosen to keep the same reliability cost (diagonally hatched area); but the homogeneous noise setting has far a higher performance cost, primarily driven by larger noise in the important synapse, $w_1$.

(i.e. allowing the noise to vary on a per-synapse basis) improved accuracy considerably for a fixed value of $c$, but only reduced the average noise slightly.

The findings in **Figure 2** imply a tradeoff between accuracy and average noise level, $\sigma$, as we change $c$. If we explicitly plot the accuracy against the noise level using the data from **Figure 2**, we see that as the synaptic noise level increases, the accuracy decreases (**Figure 3a**). Further, the synaptic noise level is associated with a reliability cost (**Figure 3b**), and this relationship changes in the different columns as they use different values of $\rho$ associated with different biological mechanisms that might give rise to the dominant biophysical reliability cost. Thus, there is also a relationship between accuracy and reliability costs (**Figure 3c**), with accuracy increasing as we allow the system to invest more energy in becoming more reliable, which implies a higher reliability cost. Again, we plotted both the homogeneous (grey lines) and heterogeneous noise cases (green lines). We found that heterogeneous noise allowed for considerably improved accuracy at a given average noise standard deviation or a given reliability cost.

## Energy-efficient patterns of synapse variability

We found that the heterogeneous noise setting, where we individually optimise synaptic noise on a per-synapse basis, performed considerably better than the homogeneous noise setting (**Figure 3**). This raised an important question: how does the network achieve such large improvements by optimising the noise levels on a per-synapse basis? We hypothesised that the system invests a lot of energy in improving the reliability for 'important' synapses, i.e., synapses whose weights have a large impact on predictions and accuracy (**Figure 4a**). Conversely, the system allows unimportant synapses to have high variability, which reduces reliability costs (**Figure 4b**). To get further intuition, we compared both $w_1$ and $w_2$ on the same plot (**Figure 4c**). Specifically, we put the important synapse, $w_1$ from **Figure 4a**, on the horizontal axis, and the unimportant synapse, $w_2$ from **Figure 4b**, on the vertical axis. In **Figure 4c**, the relative importance of the synapse is now depicted by how the cost increases as we move away from the optimal value of the weight. Specifically, the cost increases rapidly as we

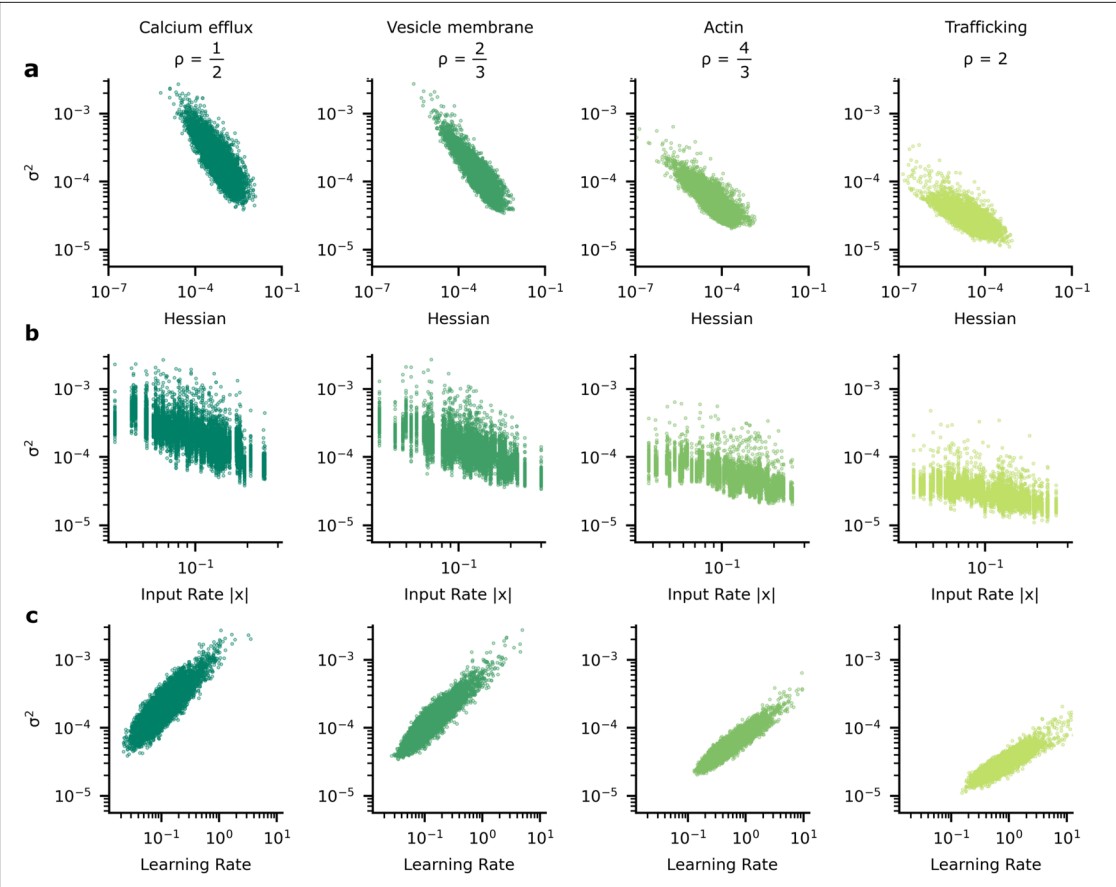

**Figure 5.** The heterogeneous patterns of synapse variability in artificial neural networks (ANNs) optimised by the tradeoff. We present data patterns on logarithmic axis between signatures of synapse importance and variability for 10,000 (100 neuron units, each with 100 synapses) synapses that connect two hidden layers in our ANN. (**a**) Synapses whose corresponding diagonal entry in the Hessian is large have smaller variance. (**b**) Synapses with high variance have faster learning rates. (**c**) As input firing rate increases, synapse variance decreases.

move away from the optimal value of $w_1$, but increases much more slowly as we move away from the optimal value of $w_2$. Now, consider deviations in the synaptic weight driven by homogeneous synaptic variability (*Figure 4c* left, grey points). Many of these points have poor performance (i.e. a high performance cost), due to relatively high noise on the important synapse (i.e. $w_1$). Next, consider deviations in the synaptic weight driven by heterogeneous, optimised variability (*Figure 4c* left, green points). Critically, optimising synaptic noise reduces variability for the important synapse, and that reduces the average performance cost by eliminating large deviations on the important synapse. Thus, for the same overall reliability cost, heterogeneous, optimised variability can achieve much lower performance costs, and hence much lower overall costs than homogeneous variability (*Figure 4d*).

To investigate experimental predictions arising from optimised, heterogeneous variability, we needed a way to formally assess the 'importance' of synapses. We used the 'curvature' of the performance cost: namely the degree to which small deviations in the weights from their optimal values will degrade performance. If the curvature is large (*Figure 4a*), then small deviations in the weights, e.g., those caused by noise, can drastically reduce performance. In contrast, if the curvature is smaller (*Figure 4b*), then small deviations in the weights cause a much smaller reduction in performance. As a formal measure of the curvature of the objective, we used the Hessian matrix, $\boldsymbol{H}$. This describes the shape of the objective as a function of the synaptic weights, the $w_i$s: specifically, it is the matrix of second derivatives of the objective, with respect to the weights, and measures the local curvature of objective. We were interested in the diagonal elements, $H_{ii}$; the second derivatives of the objective with respect to $w_i$.

We began by looking at how the optimised synaptic noise varied with synapse importance, as measured by the curvature or, more formally, the Hessian (*Figure 5a*). The Hessian values were

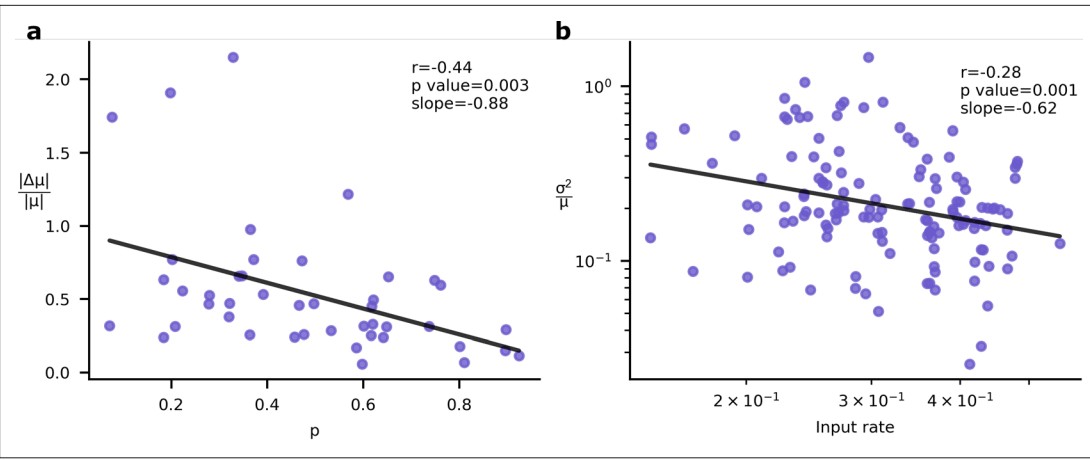

**Figure 6.** Experimental signatures of Bayesian synapses. The Bayesian synapse hypothesis predicts relationships between synapse reliability, learning rate, and input rate. (**a**) Synapses with higher probability of release, **p**, demonstrate smaller increases in synaptic mean following long-term plasticity (LTP) induction. This pattern was originally observed by **Schug et al., 2021**. (**b**) As input firing rates are increased, normalised EPSP variability decreases with $slope = -0.62$ (**Aitchison et al., 2021**).

estimated using the average-squared gradient, see Appendix 3, 'Synapse importance and gradient magnitudes'. We found that as the importance of the synapse increased, the optimised noise level decreased. These patterns of synapse variability make sense because noise is more detrimental at important synapses and so it is worth investing energy to reduce the noise in those synapses.

However, this relationship (**Figure 5a**) between the importance of a synapse and the synaptic variability is not experimentally testable, as we are not able to directly measure synapse importance. That said, we are able to obtain two testable predictions. First, the input rate in our simulations was negatively correlated with optimised synaptic variability (**Figure 5b**). Second, the optimised synaptic variability was larger for synapses with larger learning rates (**Figure 5c**). Critically, similar patterns have been observed in experimental data. In **Figure 6a** we present the negative correlation between learning rate and synaptic reliability presented by **Schug et al., 2021**, from in vitro measurements of V1 (layer 5) pyramidal synapses before and after STDP-induced long-term plasticity (LTP) conducted by **Sjöström et al., 2001**. Furthermore, a relationship between input firing rate and synaptic variability was observed by **Aitchison et al., 2021**, using in vivo functional recordings from V1 (layer 2/3) (**Ko et al., 2013**; **Figure 6b**).

To understand why these patterns of variability emerge in our simulations and in data, we need to understand the connection between synapse importance, synaptic inputs (**Figure 5b**, **Figure 6b**), and synaptic learning (**Figure 5c**, **Figure 6a**). Perhaps the easiest connection is between the synapse importance and the input firing rate. If the input cell never fires, then the synaptic weight cannot affect the network output, and the synapse has zero importance (and also zero Hessian; see Appendix 2, 'High input rates and high precision at important synapses'). This would suggest a tendency for synapses with higher input firing rates to be more important, and hence to have lower variability. This pattern is indeed borne out in our simulations (**Figure 5b**; also see **Appendix 6—figure 1**), though of course there is a considerable amount of noise: there are a few important synapses with low input rates, and vice versa.

Next, we consider the connection between learning rate and synapse importance. To understand this connection, we need to choose a specific scheme for modulating the learning rate as a function of the inputs. While the specific scheme for modulating the learning rate is ultimately an assumption, we believe modern deep learning offers strong guidance as to the optimal family of schemes for modulating the learning rate. In particular, modern, state-of-the-art, update rules for ANNs almost always use an adaptive learning rate. These adaptive learning rates, $\eta_i$ (including the most common such as Adam and variants), almost always use a normalising learning rate which decreases in response to high incoming gradients,

$$\eta_i = \frac{\eta_{\text{base}}}{\sqrt{\langle g_i^2 \rangle}}. \tag{7}$$

Specifically, the local learning rate for the $i$th synapse, $\eta_i$, is usually a base learning rate, $\eta_{\text{base}}$, divided by the root-mean-squared gradient at this synapse $\sqrt{\langle g_i^2 \rangle}$. Critically, the root-mean-squared gradient turns out to be strongly related to synapse importance. Intuitively, important synapses with greater impact on network predictions will have larger gradients (see Appendix 3, 'Synapse importance and gradient magnitudes').

In vivo performance requires selective formation, stabilisation, and elimination of LTP (*Yang et al., 2009*), raising the questions as to which biological mechanisms are able to provide this selectivity. Reducing updates at historically important synapses is one potential approach to determining which synapses should have their strengths adjusted and which should be stabilised. Adjusting learning rates based on synapse importance enables fast, stable learning (*LeCun et al., 2002*; *Kingma and Ba, 2014*; *Khan et al., 2018*; *Aitchison, 2020*; *Martens, 2020*; *Jegminat et al., 2022*).

For our purposes, the crucial point is that when training using an adaptive learning rate such as *Equation 7*, important synapses have higher root-mean-squared gradients, and hence lower learning rates. Here, we use a specific set of update rules which uses this adaptive learning rate (i.e. Adam: *Kingma and Ba, 2014*; *Yang and Li, 2022*). Thus, we can use learning rate as a proxy for importance, allowing us to obtain the predictions tested in *Figure 5b* which match *Figure 5a/c*.

## The connection to Bayesian inference

Surprisingly, our experimental predictions obtained for optimised, heterogeneous synaptic variability (*Figure 5*) match those arising from Bayesian synapses presented in *Figure 6* (i.e. synapses that use Bayes to infer their weights; *Aitchison et al., 2021*). Our first prediction was that lower variability implies a lower learning rate. The same prediction also arises if we consider Bayesian synapses. In particular, if variability and hence uncertainty is low, then a Bayesian synapse is very certain that it is close to the optimal value. In that case, new information should have less impact on the synaptic weight, and the learning rate should be lower. Our second prediction was that higher presynaptic firing rates imply less variability. Again, this arises in Bayesian synapses: Bayesian synapses should become more certain and less variable if the presynaptic cell fires more frequently. Every time the presynaptic cell fires, the synapse gets a feedback signal which gives a small amount of information about the right value for that synaptic weight. So the more times the presynaptic cell fires, the more information the synapse receives, and the more certain it becomes.

This match between observations for our energy-efficient synapses and previous work on Bayesian synapses led us to investigate potential connections between energy efficiency and Bayesian inference. Intuitively, there turns out to be a strong connection between synapse importance and uncertainty. Specifically, if a synapse is very important, then the performance cost changes dramatically when there are errors in that synaptic weight. That synapse therefore receives large gradients, and hence strong information about the correct value, rapidly reducing uncertainty.

To assess the connection between Bayesian posteriors and energy-efficient variability in more depth, we estimated and plotted the posterior variance against the optimised synaptic variability (*Figure 7a*) (see Materials and methods). We considered our four different biophysical mechanisms (values for $\rho$; *Figure 7a*, columns), and values for $c$ (*Figure 7a*, rows). In all cases, there was a clear correlation between the posterior and the optimised variability: synapses with larger posterior variance also had large optimised variance. To further assess this connection, we used the relation between the Hessian and posterior variance given by *Equation 54c* and the analytic result given in Appendix 5, 'Analytic predictions for $\sigma_i$' to plot the relationships between $\sigma_i$ and the posterior variability, $\sigma_{\text{post}}$, as a function of $\rho$ (*Figure 7b*) and as a function of $c$ (*Figure 7c*). Again, these plots show a clear correlation between synapse variance and posterior variance, though the relationship is far from perfect. For a perfect relationship, we would expect the lines in *Figure 7b and c* to all lie along the diagonal with slope equal to one. In contrast, these lines actually have a slope smaller than one, indicating that optimised variability is less heterogeneous than posterior variance (*Figure 7b and c*). Interestingly, the slope increases towards one as the associated $\rho$ is decreased, this suggests that synapse variability best approximates the posterior when $\rho$ is small.

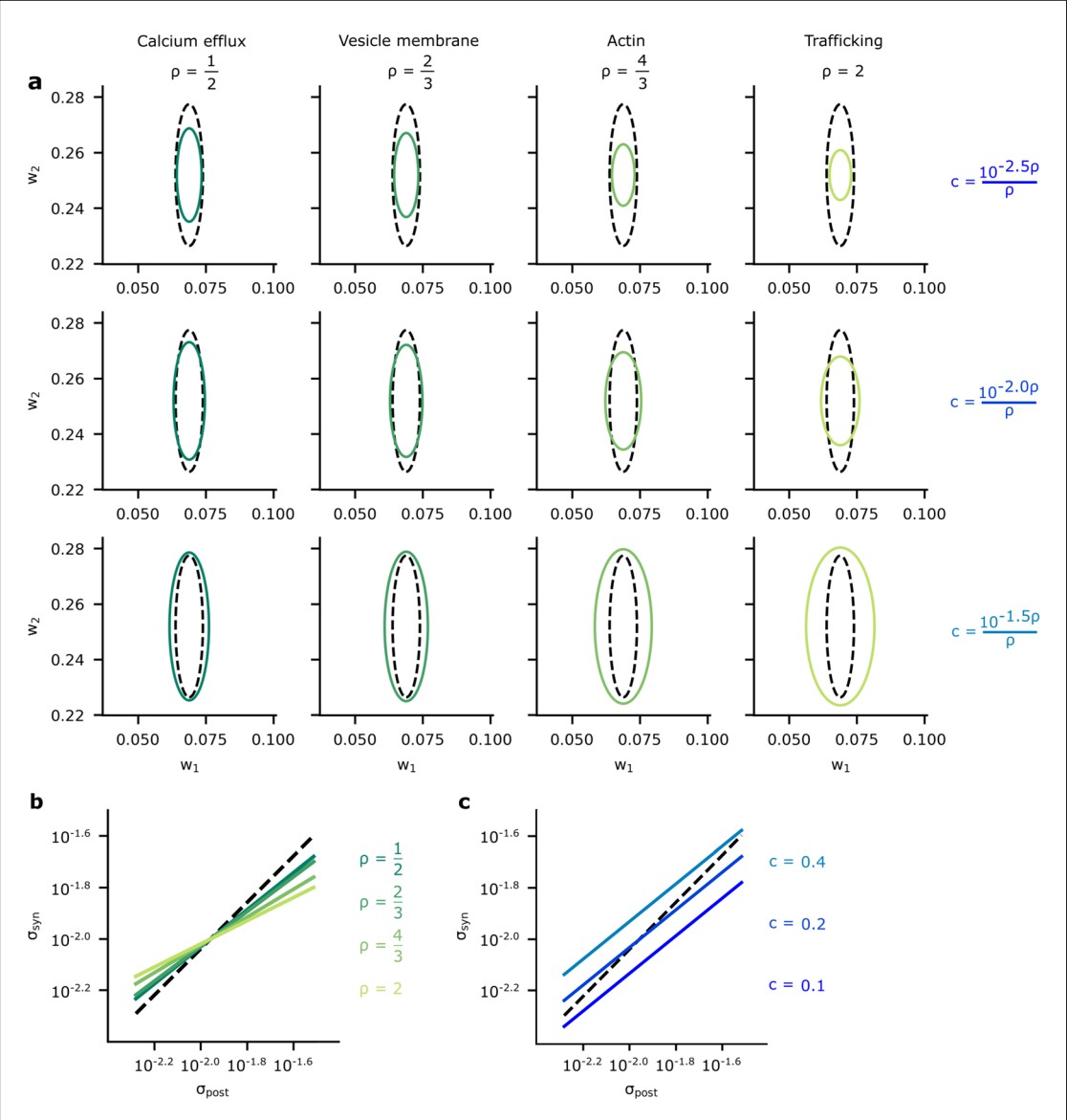

**Figure 7.** A comparison of optimised synaptic variability and posterior variance. (**a**) Posterior variance (grey-dashed ellipses) plotted alongside optimised synaptic variability (green ellipses) for different values of $\rho$ (columns) and $c$ (rows) for an illustrative pair of synapses. Note that using fixed values of $c$ for different $\rho$'s dramatically changed the scale of the ellipses. Instead, we chose $c$ as a function of $\rho$ to ensure that the scale of the optimised noise variance was roughly equal across different $\rho$. This allowed us to highlight the key pattern: that smaller values for $\rho$ give optimised variance closer to the true posterior variances, while higher values for $\rho$ tended to make the optimised synaptic variability more isotropic. (**b**) To understand this pattern more formally, we plotted the synaptic variability as a function of the posterior variance for different values of $\rho$. Note that we set $c$ to $c = \frac{10^{-2.0\rho}}{\rho}$ to avoid large additive offsets (see Connecting the entropy and the biological reliability cost – **Equation 48** for details). (**c**) The synaptic variability as a function of the posterior variance for different values of $c$ : $[0, 112, 0.2, 0.356]$ (3 DP). As $c$ increases (lighter blues) we penalise reliability more, and hence the optimised synaptic noise variability increases. (Here, we fixed $\rho = 1/2$ across different settings for $c$.)

This strong, but not perfect, connection between the patterns of variability in Bayesian inference and energy-efficient networks motivated us to seek a formal connection between Bayesian and efficient synapses. As such, in the Appendix, we derive a theoretical connection between our overall performance cost and Bayesian inference (see Appendix 4, 'Energy-efficient noise and variational Bayes for neural network weights'). Moreover, this connection is subsequently used to provide an explanation for why synapse variability aligns closer to posterior variance for small $\rho$ (see **Equation 51**), specifically, variational inference, a well-known procedure for performing (approximate) Bayesian inference in NNs (**Hinton and van Camp, 1993**; **Graves, 2011**; **Blundell et al., 2015**). Variational

inference optimises the 'evidence lower bound objective' (ELBO) (*Barber and Bishop, 1998*; *Jordan et al., 1999*; *Blei et al., 2017*), which surprisingly turns out to resemble our performance cost. Specifically, the ELBO includes a term which encourages the entropy of the approximating posterior distribution (which could be interpreted as our noise distribution) to be larger. This resembles a reliability cost, as our reliability costs also encourage the noise distribution to be larger. Critically, the biological power-law reliability cost has a different form from the ideal, entropic reliability cost. However, we are able to derive a formal relationship: the biological power-law reliability costs bound the ideal entropic reliability cost. Remarkably, this implies that our overall cost (*Equation 2*) bounds the ELBO, so reducing our cost (*Equation 2*) tightens the ELBO bound and gives an improved guarantee on the quality of Bayesian inference.

## Discussion

Comparing the brain's computational roles with associated energetic costs provides a useful means for deducing properties of efficient neurophysiology. Here, we applied this approach to PSP variability. We began by looking at the biophysical mechanisms of synaptic transmission, and how the energy costs for transmission might vary with synaptic reliability. We modified a standard ANN to incorporate unreliable synapses and trained this on a classification task using an objective that combined classification accuracy and an energetic cost on reliability. This led to a performance-reliability cost tradeoff and heterogeneous patterns of synapse variability that correlated with input rate and learning rate. We noted that these patterns of variability have been previously observed in data (see *Figure 6*). Remarkably, these are also the patterns of variability predicted by Bayesian synapses (*Aitchison et al., 2021*) (i.e. when distributions over synaptic weights correspond with the Bayesian posterior). Finally, we showed empirical and formal connections between the synaptic variability implied by Bayesian synapses and our performance-reliability cost tradeoff.

The reliability cost in terms of the synaptic variability (*Equation 4*) is a critical component of the numerical experiments we present here. While the precise form of the cost is inevitably uncertain, we attempted to mitigate the uncertainty by considering a wide range of functional forms for the reliability cost. In particular, we considered four biophysical mechanisms, corresponding to four power-law exponents, ($\rho = \frac{1}{2}, \frac{2}{3}, \frac{4}{3}, 2$). Moreover, these different power-law costs already cover a reasonably wide-range of potential penalties and we would expect the results to hold for many other forms of reliability cost as the intuition behind the results ultimately relies merely on there being *some* penalty for increasing reliability.

The biophysical cost also includes a multiplicative factor, $c$, which sets the magnitude of the reliability cost. In fact, the patterns of variability exhibited in *Figure 5* are preserved as $c$ is changed: this was demonstrated for values of $c$ which are 10 times larger and 10 times smaller, *Appendix 6—figure 2*. This multiplicative factor should be understood as being determined by the properties of the physics and chemistry underpinning synaptic dynamics, e.g., it could represent the quantity of ATP required by the metabolic costs of synaptic transmission (although this factor could vary, e.g. in different cell types).

Our ANNs used backpropagation to optimise the mean and variance of synaptic weights. While there are a number of schemes by which biological circuits might implement backpropagation (*Whittington and Bogacz, 2017*; *Sacramento et al., 2018*; *Richards and Lillicrap, 2019*), it is not yet clear whether backpropagation is implemented by the brain (see *Lillicrap et al., 2020*, for a review on the plausibility of propagation in the brain). Regardless, backpropagation is merely the route we used in our ANN setting to reach an energy-efficient configuration. The patterns we have observed are characteristic of an energy-efficient network and therefore should not depend on the learning rule that the brain uses to achieve energy efficiency.

Our results in ANNs used MNIST classification as an example of a task; this may appear somewhat artificial, but all brain areas ultimately do have a task: to maximise fitness (or reward as a proxy for fitness). Moreover, our results all ultimately arise from trading off biophysical reliability costs against the fact that if a synapse is important to performing a task, then variability in that synapse substantially impairs performance. Of course performance, in different brain areas, might mean reward, fitness, or some other measures. In contrast, if a synapse is unimportant, variability in that synapse impairs performance less. In all tasks there will be some synapses that are more, and some synapses that are less important, and our task, while relatively straightforward, captures this important property.

Our results have important implications for the understanding of Bayesian inference in synapses. In particular, we show that energy efficiency considerations give rise to two phenomena that are consistent with predictions outlined in previous work on Bayesian synapses (*Aitchison et al., 2021*). First, that normalised variability decreases for synapses with higher presynaptic firing rates. Second, that synaptic plasticity is higher for synapses with higher variability.

Specifically, these findings suggest that synapses connect their uncertainty in the value of the optimal synaptic weight (see *Aitchison et al., 2021*, for details) to variability. This is in essence a synaptic variant of the 'sampling hypothesis'. Under the sampling hypothesis, neural activity is believed to represent a potential state of the world, and variability is believed to represent uncertainty (*Hoyer and Hyvärinen, 2002*; *Knill and Pouget, 2004*; *Ma et al., 2006*; *Fiser et al., 2010*; *Berkes et al., 2011*; *Orbán et al., 2016*; *Aitchison and Lengyel, 2016*; *Haefner et al., 2016*; *Lange and Haefner, 2017*; *Shivkumar et al., 2018*; *Bondy et al., 2018*; *Echeveste et al., 2020*; *Festa et al., 2021*; *Lange et al., 2021*; *Lange and Haefner, 2022*). This variability in neural activity, representing uncertainty in the state of the world, can then be read out by downstream circuits to inform behaviour. Here, we showed that a connection between synaptic uncertainty and variability can emerge simply as a consequence of maximising energy efficiency. This suggest that Bayesian synapses may emerge without any necessity for specific synaptic biophysical implementations of Bayesian inference.

Importantly though, while the brain might use synaptic noise for Bayesian computation, these results are also consistent with an alternative interpretation: that the brain is not Bayesian, it just looks Bayesian because it is energy efficient. To distinguish between these two interpretations, we ultimately need to know whether downstream brain areas exploit or ignore information about uncertainty that arises from synaptic variability.

## Materials and methods

The ANN simulations were run in PyTorch with feedforward, fully connected neural networks with two hidden layers of width 100. The input dimension of 784 corresponded to the number of pixels in the greyscale MNIST images of handwritten digits, while the output dimension of 10 corresponded to the number of classes. We used the reparameterisation trick to backpropagate with respect to the mean and variance of the weights, in particular, we set $w_i = \mu_i + \sigma_i \xi$, where $\xi \sim \mathrm{Normal}(0, 1)$ (*Kingma et al., 2015*). MNIST classification was learned through optimisation of Gaussian parameters with respect to a cross-entropy loss in addition to reliability costs using minibatch gradient descent under Adam optimisation with a minibatch size of 20. To prevent negative values for the $\sigma$s, they were reparameterised using a softplus function with argument $\phi_i$, with $\sigma_i = \mathrm{softplus}(\phi_i)$. The base learning rate in *Equation 7* is $\eta_{\mathrm{base}} = 5 \times 10^{-4}$. The $\mu_i$s were initialised homogeneously across the network from $\mathrm{Uniform}(-0.1, 0.1)$ and the $\sigma_i$s were initialised homogeneously across the network at $10^{-4}$. Hyperparameters were chosen via grid search on the validation dataset to enable smooth learning, high performance, and rapid convergence. In the objective $\mathcal{L}_{\mathrm{BI}}$ used to train our simulations, we also add an L1 regularisation term over synaptic weights, $\lambda |\mu|_1$, where $\lambda = 10^{-4}$.

Plots in *Figure 2* present mappings from hyperparameter, $c$, to accuracy and $\sigma$. A different neural network was trained for each $c$, after 50 training epochs the average $\sigma$ across synapses was computed, and accuracy was evaluated on the test dataset. Plots in *Figure 3* present mappings of this $\sigma$ against accuracy and reliability cost. The reliability cost was computed using fixed $s = 1$ (see *Equation _48*).

To compute the Hessian in *Figure 5* and elsewhere, we used the empirical Fisher information approximation (*Fisher, 1922*), $H \approx g^2$. This was evaluated by taking the average $g^2$ at $w^* = \mu$ over 10 epochs after full training for 50 epochs. The average learning rate $\gamma |g|^{-1}$ and the average input rate $|x|$ were also evaluated over 10 epochs following training. The data presented illustrate these variables with regard to the weights of the second hidden layer. We set hyperparameter $s = 0.001$ (see *Equation 47*) in these simulations.

For geometric comparisons between the distribution over synapses and the Bayesian posterior presented in *Figure 7* we used the analytic results in Appendix 5, 'Analytic predictions for $\sigma_i$'. To estimate the posterior used in *Figure 7*, we optimised a factorised Gaussian approximation to the posterior over weights using variational inference and Bayes by backpropagation (*Blundell et al., 2015*). We then took $\sigma_{post}$ from two optimised weights. For the variance and slope comparisons between Bayesian and efficient synapses in *Figure 7*, we used the analytic results in Appendix 5, 'Analytic predictions for $\sigma_i$'.

Source code used in simulations is available at: https://github.com/JamesMalkin/EfficientBayes copy archived at *Malkin, 2024*.

## Acknowledgements

We are grateful to Dr Stewart whose philanthropy supported GPU compute used in this project. JM was funded by the Engineering and Physical Sciences Research Council (EP/T517872/1). COD was funded by the Leverhulme Trust (RPG-2019-229) and Biotechnology and Biological Sciences Research Council (BB/W001845/1). CH is supported by the Leverhulme Trust (RF-2021-533).

## Additional information

### Funding

| Funder | Grant reference number | Author |
|---|---|---|
| Engineering and Physical Sciences Research Council | EP/T517872/1 | James Malkin |
| Leverhulme Trust | RPG-2019-229 | Cian O'Donnell |
| Biotechnology and Biological Sciences Research Council | BB/W001845/1 | Cian O'Donnell |
| Leverhulme Trust | RF-2021-533 | Conor J Houghton |

The funders had no role in study design, data collection and interpretation, or the decision to submit the work for publication.

### Author contributions

James Malkin, Conceptualization, Formal analysis, Investigation, Visualization, Methodology, Writing - original draft, Writing – review and editing; Cian O'Donnell, Supervision, Funding acquisition, Visualization, Writing – review and editing; Conor J Houghton, Formal analysis, Supervision, Visualization, Writing – review and editing; Laurence Aitchison, Conceptualization, Formal analysis, Supervision, Visualization, Methodology, Writing – review and editing

### Author ORCIDs

James Malkin ⬥ https://orcid.org/0009-0001-7473-1442
Cian O'Donnell ⬥ https://orcid.org/0000-0003-2031-9177
Conor J Houghton ⬥ https://orcid.org/0000-0001-5017-9473

Reviewer #1 (Public Review): https://doi.org/10.7554/eLife.92595.3.sa1
Reviewer #2 (Public Review): https://doi.org/10.7554/eLife.92595.3.sa2
Author response https://doi.org/10.7554/eLife.92595.3.sa3

## Additional files

### Supplementary files
• MDAR checklist

### Data availability
The current manuscript is a computational study, so any data generated is simulated data. The previously published dataset listed below and data from *Ko et al., 2013* were used. All newly generated data is available at: https://github.com/JamesMalkin/EfficientBayes, copy archived at *Malkin, 2024*.

The following previously published dataset was used:

| Author(s) | Year | Dataset title | Dataset URL | Database and Identifier |
|---|---|---|---|---|
| Costa R, Froemke R, Sjöström P, van Rossum M | 2015 | Data from: Unified pre- and postsynaptic long-term plasticity enables reliable and flexible learning | https://doi.org/10.5061/dryad.p286g | Dryad Digital Repository, 10.5061/dryad.p286g |

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

# Appendix 1

## Reliability costs

The difficulty in determining reliability costs is that $\sigma$ depends on three variables: $n$, the number of vesicles, $p$ the probability of release, and $q$, the quantal size, which measures the amount of neurotransmitter in each vesicle:

$$\sigma^2 = np(1-p)q^2. \tag{8}$$

However, these variables also determine the mean

$$\mu = npq \tag{9}$$

so a straightforward optimisation of $\sigma$ under reliability costs will also change $\mu$ and one pitfall is to accidentally consider only those changes in reliability that derive from changes in mean. A solution to this problem is to eliminate one of the variables so that $\sigma$ is a function of $\mu$, and the remaining variables. Eliminating $q$ gives

$$\sigma = \mu\sqrt{\frac{1-p}{np}}. \tag{10}$$

The idea is to consider energetic costs associated with $p$ and $n$ and relate these to $\sigma$ while holding $\mu$ fixed. To simplify the biological motivation, we assume that during changes to the synapse aimed at manipulating the energetic cost, $q$ will also change to compensate for any collateral changes in $\mu$ keeping $\mu$ constant. Hence, $q$ a 'compensatory variable'. Moreover, there is biological evidence that $q$ is the mechanism used in real synapses to fix $\mu$ (*Turrigiano et al., 1998*; *Karunanithi et al., 2002*). Fixing $\mu$ through a compensatory variable is termed homeostatic plasticity. Fixing $\mu$ through $q$ is described as 'quantal scaling'. For reviews on homeostatic plasticity, see *Turrigiano and Nelson, 2004*; *Davis and Müller, 2015*.

In what follows four different energy costs are considered, the first depends on $p$, the next two on $n$, the situation for the final one is less clear, but in each case we derive a reliability cost in the form $\sigma^{-\rho}$ for some value of $\rho$. Of course, since we are considering costs for fixed $\mu$ the coefficient of variation $k = \sigma/\mu$ is proportional to $\sigma$; it may be helpful to think of these calculations as finding the relationship between the cost and $k$.

## Calcium efflux – $\rho = \frac{1}{2}$

Calcium influx into the synapse is an essential part of the mechanism for vesicle release. Presynaptic calcium pumps act to restore calcium concentrations in the synapse; this pumping is a significant portion of synaptic transmission costs (*Attwell and Laughlin, 2001*). By rearranging the Hill equation defined by *Sakaba and Neher, 2001*, it can be shown that vesicle release has an interaction coefficient of four, this means the *odds* of release per vesicle are related to intracellular calcium amplitude via the fourth power (*Heidelberger et al., 1994*; *Sakaba and Neher, 2001*):

$$\frac{p}{1-p} \propto [Ca]^4. \tag{11}$$

To recover basal synaptic calcium concentration, the calcium influx is reversed by ATP-driven calcium pumps, where $[Ca] \propto ATP$:

$$\text{Calcium pump cost} \propto \sqrt[4]{\frac{p}{1-p}}. \tag{12}$$

Since this physiological cost does not depend on $n$ we assume it is fixed and so the odds of release $p/(1-p)$ is proportional to $\sigma^{-2}$. Thus,

$$\text{Calcium pump cost} \propto \frac{1}{\sigma^{\frac{1}{2}}} \tag{13}$$

or $\rho = 1/2$.

## Vesicle membrane – $\rho = \frac{2}{3}$

There is a cost associated with the total area of vesicle membrane. Evidence in **Pulido and Ryan, 2021**, suggest stored vesicles emit charged H⁺ ions, with the number emitted proportional to the number of v-glut, glutamate transporters, on the surface of vesicles. v-ATPase pumps reverse this process maintaining the pH of the cell. It is suggested that this cost is 44% of the resting synaptic energy consumption. In addition, metabolism of the phospholipid bilayer that form the membrane of neurotransmitter-filled vesicles has been identified as a major energetic cost (**Purdon et al., 2002**). Provided the total volume is the same, release of the same amount of neurotransmitter into the synaptic cleft can involve many smaller vesicles or fewer larger ones. However, while having many small vesicles will be more reliable, it requires a greater surface area of costly membrane. With fixed $\mu$ and $p$,

$$\text{Vesicle membrane cost} \propto nr^2. \tag{14}$$

Since $r^2 \propto q^{2/3}$ and using $\mu = npq$ this give

$$\text{Vesicle membrane cost} \propto \left(\frac{n\mu^2}{p^2}\right)^{1/3}. \tag{15}$$

Since this reliability cost depends on $n$, so $p$ is regarded as constant, so $\sigma^{-2} \propto n$ and hence $\rho = 2/3$.

## Actin – $\rho = \frac{4}{3}$

Actin polymers are an energy costly structural filament that support the structural organisation of vesicle reserve pools (**Cingolani and Goda, 2008**), with the vesicles strung out along the filaments. We assume each vesicle to require a length of actin roughly proportional to its diameter, this means that the total length of actin is proportional to $nr$. Hence,

$$\text{Actin cost} \propto nr. \tag{16}$$

The calculation then proceeds much as for the membrane cost, but with $r$ instead of $r^2$ giving $\rho = 4/3$.

## Trafficking – $\rho = 2$

ATP-fuelled myosin motors drive trains of actin filament along with associated cargo such as vesicles and actin-myosin trafficking moves vesicles from vesicle reserve pools to release sites sustaining the readily releasable pool following vesicle release (**Bridgman, 1999**; **Gramlich and Klyachko, 2017**). This gives a cost for vesicle recruitment proportional to $np$, the number of vesicles released:

$$\text{Trafficking cost} \propto np \tag{17}$$

so if $n$ is regarded as the principal way this cost is changed, with $p$ fixed then $n \propto \sigma^{-2}$ and so $\rho = 2$. This is the view point we are taking to motivate examining reliability costs with $\rho = 2$. This is certainly useful in considering the range of model behaviours over a broad range of $\rho$ values.

Nonetheless, it is sensible to ask whether the likely biological mechanism behind a varying trafficking cost is one which changes $np$ itself. In this case, since a constant $\mu$ for varying $np$ means $q \propto 1/np$, we have

$$\text{Trafficking cost} \propto \frac{1-p}{\sigma^2} \tag{18}$$

which, is, again, of the form cost $\propto \sigma^{-2}$ provided $p$ is small. For larger $p$, however, it depends on exactly how $p$ changes as $np$ changes.

Generally, throughout these calculation we have supposed that $q$ is a compensatory variable and that either $p$ or $n$ changes in the process that changes the cost at a synapse. The benefit of this is that we are able to model costs directly in terms of reliability; in the future, though, it might be interesting to consider models which use $n$, $p$, and $q$ instead of $\mu$ and $\sigma$; this would certainly be convenient for the sort of comparisons we are making here and interesting, although two formulations seem equivalent, this does not mean that the learning dynamics will be the same.

## Appendix 2

### High input rates and high precision at important synapses

Here, we show that under a linear model, important synapses, as measured by the Hessian, have high input rates. Additionally, we show that energy-efficient synapses imply that these important synapses have low optimized variability.

We consider a simplified linear model, with targets $\boldsymbol{y}$, inputs $\boldsymbol{X}$, and weights $\boldsymbol{w}$. We have

$$P(\boldsymbol{y}|\boldsymbol{w}, \boldsymbol{X}) = \mathcal{N}(\boldsymbol{y}; \boldsymbol{Xw}, \epsilon^2 \boldsymbol{I}). \tag{19}$$

We take the performance cost to be

$$\text{performance cost} = -\log P(\boldsymbol{y}|\boldsymbol{w}, \boldsymbol{X}) \tag{20}$$

where $N$ is the number of datapoints. We take the network weights to be drawn from a multivariate Gaussian, with diagonal covariance, $\boldsymbol{\Sigma}$, i.e., $\Sigma_{ii} = \sigma_i^2$,

$$Q(\boldsymbol{w}) = \mathcal{N}(\boldsymbol{w}; \boldsymbol{\mu}, \boldsymbol{\Sigma}). \tag{21}$$

All our derivations rely on looking at the quadratic form for the performance cost. In particular,

$$\text{performance cost} = \mathrm{E}_{Q(\boldsymbol{w})}\left[\frac{1}{2\epsilon^2}(\boldsymbol{y} - \boldsymbol{Xw})^T(\boldsymbol{y} - \boldsymbol{Xw})\right]. \tag{22}$$

Expanding the brackets,

$$\text{performance cost} = \frac{1}{2\epsilon^2}(\boldsymbol{y}^T\boldsymbol{y} - 2\boldsymbol{y}^T\boldsymbol{X}[w] + E[\boldsymbol{w}^T\boldsymbol{X}^T\boldsymbol{Xw}]), \tag{23}$$

and evaluating the expectations,

$$\text{performance cost} = \frac{1}{2\epsilon^2}[\boldsymbol{yy}^T - 2\boldsymbol{y}^T\boldsymbol{X\mu} + \mathrm{tr}(\boldsymbol{X}^T\boldsymbol{X\Sigma}) + \boldsymbol{\mu}^T\boldsymbol{X}^T\boldsymbol{X\mu}], \tag{24}$$

where the trace,

$$\mathrm{tr}(\boldsymbol{X}^T\boldsymbol{X\Sigma}) = \sum_i \sum_t X_{ti}^2 \sigma_i^2, \tag{25}$$

includes a sum over both the synapse index $i$ and the time index $t$.

To measure synapse importance, we use the Hessian, i.e., the second derivative of the performance cost with respect to the mean. From **Equation 24**, we can identify this as

$$H_{ii} = \frac{\partial^2 \text{performance cost}}{\partial \mu_i^2} = \frac{1}{\epsilon^2}\sum_t X_{ti}^2. \tag{26}$$

where, as before, $t$ is the time index.

Since the the only term in the performance cost that depends on the variance, $\sigma_i^2$, is the trace, we can identify $\sigma_i$ that optimises the performance and reliability cost. This allows us to observe how synapse variability relates to synapse importance in the tradeoff between performance and reliability costs:

$$\frac{d}{d\sigma_i}[\text{performance cost} + \text{reliability cost}] = H_{ii}\sigma_i - s^\rho \sigma_i^{-\rho-1} = 0 \tag{27}$$

which means

$$\sigma_i^{\rho+2} = s^\rho / H_{ii} \tag{28}$$

or

$$\sigma_i = \sqrt[\rho+2]{s^\rho / H_{ii}}. \tag{29}$$

Thus, more important synapses (as measured by the Hessian, $H_{ii}$) have lower variability if the synapse is energy efficient. Moreover, through *Equation 26*, we expect synapses with higher input rates to be more important.

## Appendix 3

### Synapse importance and gradient magnitudes

In our ANN simulations we train synaptic weights using the most established adaptive optimisation scheme, Adam (*Kingma and Ba, 2014*), which has recently been realised using biologically plausible mechanisms (*Yang and Li, 2022*). Adam uses a synapse-specific learning rate, $\eta_i$, which decreases in response to high gradients at that synapse,

$$\eta_i = \frac{\eta_{\text{base}}}{\sqrt{\langle g_i^2 \rangle}}. \tag{30}$$

Specifically, the local learning rate for the $i$th synapse, $\eta_i$, is usually a base learning rate, $\eta_{\text{base}}$, divided by $\sqrt{\langle g_i^2 \rangle}$, the root-mean-square of the gradient for each datapoint or minibatch.

The key intuition is that if the gradients for each datapoint/minibatch are large, that means that this synapse is important, as it has a big impact on the predictions for every datapoint. In fact, these mean-squared gradients can be related to our formal measure of synapse importance, the Hessian. Specifically, for data generated from the model, the Hessian (or Fisher information; *Fisher, 1922*) is equivalent to the mean-squared gradient, where the gradient is taken over each datapoint separately,

$$E_{P(y,\boldsymbol{x}|\boldsymbol{w})} \left[ \frac{\partial^2 \text{performance cost}}{\partial \boldsymbol{\mu}^2} \right] = -E_{P(y,\boldsymbol{x}|\boldsymbol{w})} \left[ \frac{\partial^2 \log P(y|\boldsymbol{w},\boldsymbol{x})}{\partial \boldsymbol{\mu}^2} \right] = E_{P(y,\boldsymbol{x}|\boldsymbol{w})} [\boldsymbol{g}\boldsymbol{g}^T], \tag{31}$$

where the expectation is evaluated over data generated by the model and $\boldsymbol{g}$ is defined,

$$\boldsymbol{g} = \frac{\partial}{\partial \boldsymbol{\mu}} \log P(y|\boldsymbol{w},\boldsymbol{x}). \tag{32}$$

Of course, in practice, the data is not drawn from the model, so the relationship between the squared gradients and the Hessian computed for real data is only approximate. But it is close enough to induce a relationship between synapse importance (measured as the diagonal of the Hessian) and learning rates (which are inversely proportional to the root-mean-squared gradients) in our simulations.

## Appendix 4

### Energy-efficient noise and variational Bayes for neural network weights

#### Introduction to variational Bayes for neural network weights

One approach to performing Bayesian inference for the weights of a neural network is to use variational Bayes. In variational Bayes, we introduce a parametric approximate posterior, $Q(w)$, and fit the parameters of that approximate posterior using gradient descent on an objective, the ELBO (**Blundell et al., 2015**). In particular,

$$\log P(y|x) \geq \text{ELBO} = E_{Q(w)}[\log P(y|x, w) + \log P(w)] + H[Q(w)]. \tag{33}$$

where $H[Q(w)]$ is the entropy of the approximate posterior. Maximising the ELBO is particularly useful for selecting models as it forms a lower bound on the marginal likelihood, or evidence, $\log P(y|x)$ (**MacKay, 1992b**; **Barber and Bishop, 1998**). When using variational Bayes for neural network weights, we usually use Gaussian approximate posteriors (**Blundell et al., 2015**),

$$Q(w_i) = \mathcal{N}(w_i; \mu_i, \sigma_i^2). \tag{34}$$

Note that optimising the ELBO with respect to the parameters of $Q$ is difficult, because $Q$ is the distribution over which the expectation is taken in **Equation 33**. To circumvent this issue, we use the reparameterisation trick (**Kingma et al., 2015**), which involves writing the weights in terms of IID standard Gaussian variables, $\epsilon$,

$$w_i = \mu_i + \sigma_i \epsilon_i. \tag{35}$$

Thus, we can write the ELBO as an expectation over $\epsilon$, which has a fixed IID standard Gaussian distribution,

$$\text{ELBO} = E_\epsilon \left[ \log P(y|x, w{=}\mu{+}\sigma\epsilon) + \log P(w{=}\mu{+}\sigma\epsilon) \right] + H[Q(w)]. \tag{36}$$

Note that we use $w$, $\mu$, etc. without indices in this expression to indicate all the weights/mean weights.

#### Identifying the log-likelihood and log-prior

Following the usual practice in deep learning, we assume the likelihood is formed by a categorical distribution, with probabilities obtained by applying the softmax function to the output of the neural network. The log-likelihood for a categorical distribution with softmax probabilities is the negative cross-entropy (**Murphy, 2012**),

$$\log P(y|x, w) = -\text{cross entropy}(y; f(x, w)) \tag{37}$$

where $f(x, w)$ is the output of the network with weights, $w$, and inputs, $x$. We can additionally identify a Laplace prior with the magnitude cost. Specifically, if we take the prior to be Laplace, with scale $1/\lambda$,

$$\log P(w) = \sum_i \log \text{Laplace}(w_i; 0, \tfrac{1}{\lambda}) \tag{38}$$

$$= -N \log(2/\lambda) - \lambda \sum_i |w_i| \tag{39}$$

$$= \text{const} - \text{magnitude cost} \tag{40}$$

where we identify the magnitude cost using **Equation 3**.

#### Connecting the entropy and the biological reliability cost

Now, we have identified the log-likelihood and log-prior terms in the ELBO (**Equation 33**) with terms in our biological cost (**Equation 2**). We thus have two terms left: the entropy in the ELBO (**Equation 33**) and the reliability cost in the biological cost (**Equation 2**). Critically, the entropy term also acts as a reliability cost, in that it encourages more variability in the weights (as the entropy is positive in the ELBO [**Equation 33**] and we are trying to maximise the ELBO, the entropy term gives a bonus for

more variability). Intuitively, we can think of the entropy term in the ELBO (*Equation 33*) as being an 'entropic reliability cost', as compared to the 'biological reliability cost' in *Equation 2*.

This intuitive connection suggests that we might be able to find a more formal link. Specifically, we can define an entropic reliability cost as simply the negative entropy,

$$\mathcal{C}_{\text{VI};i} = -H[Q_i] = -\tfrac{1}{2}\ln 2\pi e\sigma_i^2. \tag{41}$$

Our goal is to write the biological reliability cost $\mathcal{C}_{\text{BI};i}$, as a bound on $\mathcal{C}_{\text{BI};i}$. By rearranging and introducing $C_{BI;i}$ and $s$,

$$\mathcal{C}_{\text{VI};i} = \frac{1}{\rho}\ln\left(\frac{s}{\sigma_i}\right)^{\rho} - \tfrac{1}{2}\ln 2\pi e s^2 \tag{42}$$

and noting that $\log a \leq a - 1$,

$$\log\left(\frac{s}{\sigma}\right)^{\rho} \leq \left(\frac{s}{\sigma_i}\right)^{\rho} - 1 \tag{43}$$

we can demonstrate that any reliability cost expressed as a generic power-law forms an upper bound on the entropic cost

$$\mathcal{C}_{\text{VI};i} \leq \frac{1}{\rho}\left(\left(\frac{s}{\sigma_i}\right)^{\rho} - 1\right) - \tfrac{1}{2}\log 2\pi e s^2 = \mathcal{C}_{\text{BI};i} = \text{ reliability cost}_i + \text{const}_i.. \tag{44}$$

Here, $\mathcal{C}_{\text{BI};i}$ is the biological reliability cost for a synapse, which formally bounds the entropic reliability cost from VI,

$$\text{const}_i = -\frac{1}{\rho} - \frac{1}{2}\log 2\pi e s^2 \qquad \text{reliability cost}_i = \frac{1}{\rho}\left(\frac{s}{\sigma_i}\right)^{\rho} = c\sigma_i^{-\rho}. \tag{45}$$

We can also sum these quantities across synapses,

$$\text{const} = \sum_i \text{const}_i \qquad \text{reliability cost} = \sum_i \text{reliability cost}_i. \tag{46}$$

$$\mathcal{C}_{\text{VI}} = \sum_i \mathcal{C}_{\text{VI};i} \qquad \mathcal{C}_{\text{BI}} = \sum_i \mathcal{C}_{\text{BI};i}. \tag{47}$$

The parameter $c$ sets the importance of the reliability cost within the performance-reliability cost tradeoff (see *Figure 2*)

$$c = s^{\rho}/\rho. \tag{48}$$

Note that while both reliability cost$_i$ and $\mathcal{C}_{\text{BI};i}$ represent the biological reliability costs, they are slightly different in that $\mathcal{C}_{\text{BI};i}$ includes an additive constant. Importantly, this additive constant is independent of $\sigma_i$ and $\mu_i$, so it does not affect learning and can be ignored.

Given that the biological reliability cost forms a bound on the ideal entropic reliability cost we can consider using the biological reliability cost in place of the entropic reliability cost,

$$\log P(y|x) \geq \text{ELBO} = E_{Q(w)}[\log P(y|x, w) + \log P(w)] - \mathcal{C}_{\text{VI}}$$
$$\log P(y|x) \geq \text{ELBO} \geq E_{Q(w)}[\log P(y|x, w) + \log P(w)] - \mathcal{C}_{\text{BI}} = -\text{overall cost} - \text{const} \tag{49}$$

we find that our overall biological cost (*Equation 2*) forms a bound on the ELBO, which itself forms a bound on the evidence, $\log P(y|x)$. Thus, pushing down the overall biological cost (*Equation 2*) pushes up a bound on the model evidence.

## Predictive probabilities arising from biological reliability costs

Given the connections between our overall cost and the ELBO, we expect that optimising our overall cost will give a similar result to variational Bayes. To check this connection, we plotted the distribution of predictions induced by noisy weights arising from variational Bayes (*Appendix 4—figure 1a*) and our overall costs (*Appendix 4—figure 1b*). Variational Bayes maximises the ELBO, therefore

its predictive distribution is optimised to reflect the data distribution from which data is drawn (*MacKay, 1992a*). We found comparable patterns for the predictive distributions learned through variational Bayes and our overall costs, albeit with some breakdown in predictive performance with higher values for $\rho$.

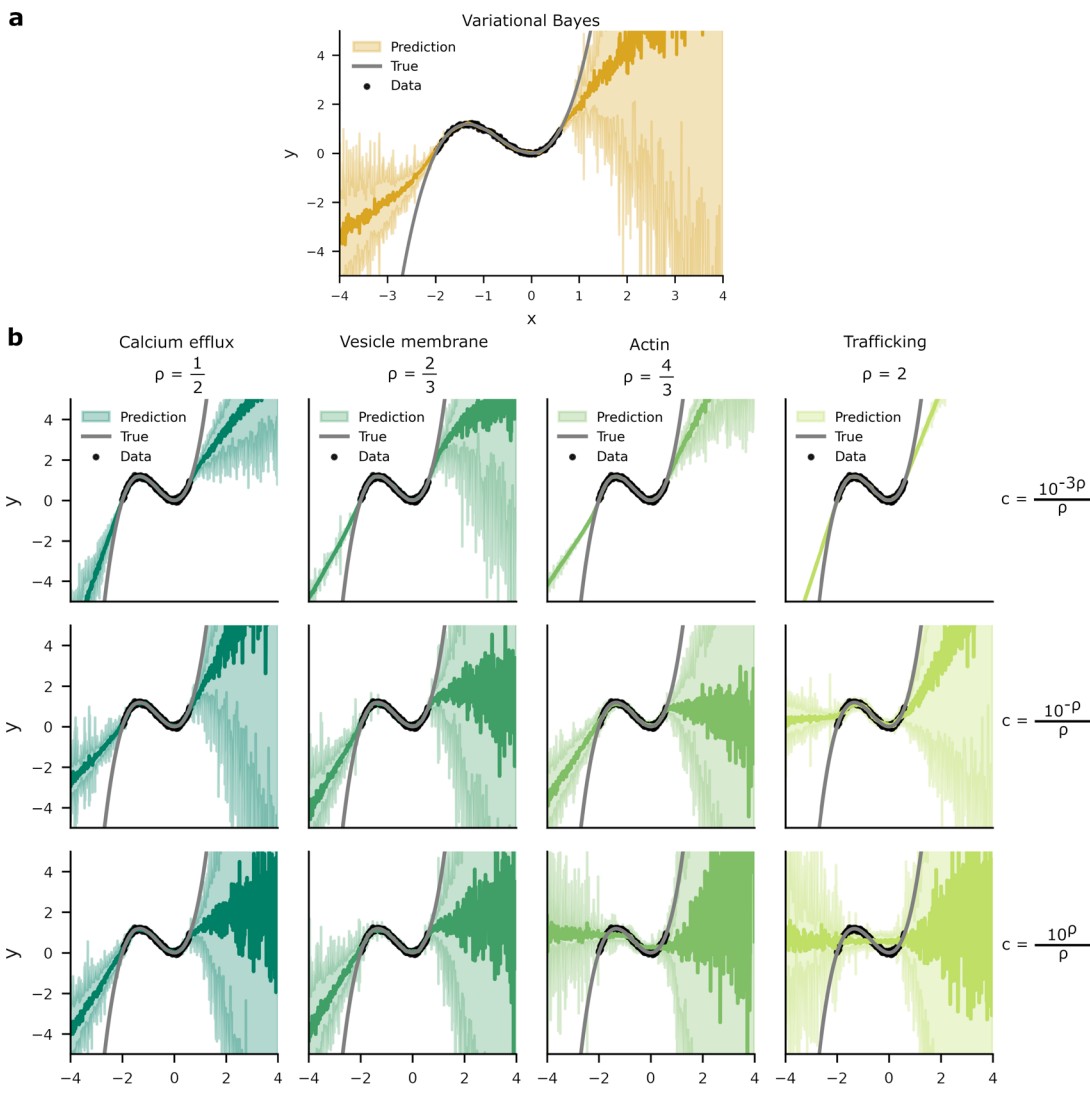

**Appendix 4—figure 1.** Predictive distributions from variational Bayes and our overall costs are similar. We trained a one hidden layer network with 20 hidden units on data $(y_a, x_a)$ (black dots). Network targets, $y_a$, were drawn from a 'true' function, $f(x) = x^3 + 2x^2$ (grey line) with additive Gaussian noise of variance $0.05^2$. Two standard deviations of the predictive distributions are depicted in the shaded areas. (**a**) The predictive distribution produced by variational Bayes have a larger density of predictions where there is a higher probability of data. Where there is an absence of data, the model has to extrapolate and the spread of the predictive distribution increases. (**b**) Optimising the overall cost with small $c$ generates narrow response distributions. This is most noticeable in the upper-right panel, where the spread of predictive distribution is unrelated to the presence or absence of data. In contrast, while the predictive density for larger $c$ do vary according to the presence or absence of data, these distributions poorly predict $f(x)$. This is most apparent in the lower-right panel where the network's predictive distribution transects the inflections of $f(x)$.

*Interpreting $c$, $\rho$, and $s$.* As discussed in the main text, $c$ and $\rho$ are the fundamental parameters, and they are set by properties of the underlying biological system. It may nonetheless be interesting to consider the effects of $s$ and $\rho$ on the tightness of the bound of the biological reliability cost on the variational reliability cost. In particular, we consider settings of $s$ and $\rho$ for which the bound is looser

or tighter (though again, there is no free choice in these parameters: they are set by properties of the biological system).

First, the biological reliability cost becomes equal to the ideal entropic reliability cost in the limit as $\rho \to 0$.

$$\lim_{x \to 0} \frac{1}{x}(z^x - 1) = \log z \tag{50}$$

Thus, taking $\rho = x$ and $z = s/\sigma_i$,

$$\lim_{\rho \to 0} \frac{1}{\rho}\left(\left(\frac{s}{\sigma_i}\right)^\rho - 1\right) = \log \frac{s}{\sigma_i} \tag{51}$$

Thus,

$$\lim_{\rho \to 0} \mathcal{C}_{\text{BI};i} = \log \frac{s}{\sigma_i} - \frac{1}{2}\log 2\pi e s^2 = -\frac{1}{2}\log 2\pi e \sigma_i^2 = \mathcal{C}_{\text{VI};i}. \tag{52}$$

This explains the apparent improvement in predictive performance (**Appendix 4—figure 1**) and in matching the posteriors (**Figure 7**) with lower values of $\rho$.

Second, the biological reliability cost becomes equal to the ideal entropic reliability cost when $s = \sigma_i$,

$$\mathrm{C}_{\text{BI};i}(s = \sigma_i) = -\frac{1}{2}\log 2\pi e s^2 = -\frac{1}{2}\log 2\pi e \sigma_i^2 = \mathrm{C}_{\text{VI};i}. \tag{53}$$

as the first term in **Equation 43** cancels. However, $s$ cannot be set individually across synapses, but is instead roughly constant, with a value set by underlying biological constraints. In particular, $s$ can be written as a function of $\rho$ and $c$ (**Equation 47**), and $\rho$ and $c$ are quantities that are roughly constant across synapses, with their values set by biological constraints. Thus, biological implications of a tightening bound as $s$ tends to $\sigma_i$ are not clear.

## Appendix 5

### Analytic predictions for $\sigma_i$

At various points in the main text, we note a connection between the Hessian, synapse importance, and optimal variability. We start with *Equation 29*, which relates the optimal, energy-efficient noise variance, $\sigma_i^2$, to the Hessian, $H_{ii}$ which gives *Equation 54a*. Then, we combine this with the form for the Hessian (*Equation 26*), which gives *Equation 54b*. Finally, we note Hessian describes the log-likelihood (*Equation 26* and *Equation 20*). Thus, assuming the prior variance is large, we have $H_{ii} = \sigma_{\text{post};i}$, which gives *Equation 54c*,

$$H_{ii} \propto \langle g_i^2 \rangle \propto \frac{1}{\eta_i^2} \qquad \log H_{ii} = -2 \log \eta_i + \text{const} \qquad \log \sigma_i^2 = \frac{4}{\rho+2} \log \eta_i + \text{const} \tag{54a}$$

$$H_{ii} \propto \langle x_i^2 \rangle \qquad \log H_{ii} = \log \langle x_i^2 \rangle \qquad \log \sigma_i^2 = -\frac{2}{\rho+2} \log \langle x_i^2 \rangle + \text{const} \tag{54b}$$

$$H_{ii} \propto \frac{1}{\sigma_{\text{post};i}^2} \qquad \log H_{ii} = -\log \sigma_{\text{post};i}^2 + \text{const} \qquad \log \sigma_i^2 = \frac{2}{\rho+2} \log \sigma_{\text{post};i}^2 + \text{const.} \tag{54c}$$

To test these predictions, we performed a simpler simulation classifying MNIST in a network with no hidden layers. We found that the analytic results closely matched the simulations, and that the biological slopes tend to match the $\rho = 2$ better than the other values for $\rho$. However, while the direction of the slope was consistent in deeper networks, the exact value of the slope was not consistent (*Figure 5* and *Appendix 6—figure 1*), so it is unclear whether we can draw any strong conclusions here.

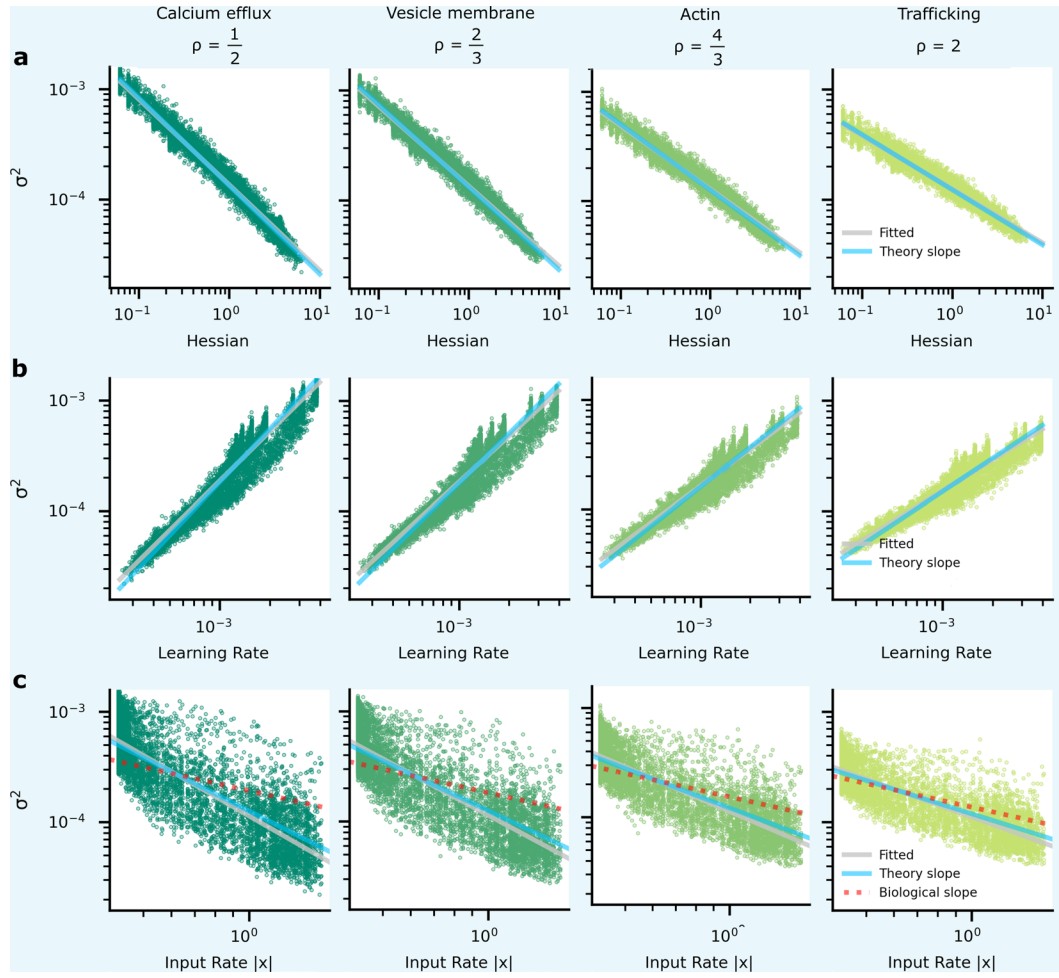

**Appendix 5—figure 1.** Comparing analytic predictions for synapse variability with simulations and experimental data in a zero hidden layer network for MNIST classification. The green dots show simulated synapses, and the

grey line is fitted to these simulated points. The blue line is from our analytic predictions, while the red-dashed line is taken from experimental data (*Figure 6b*).

# Appendix 6

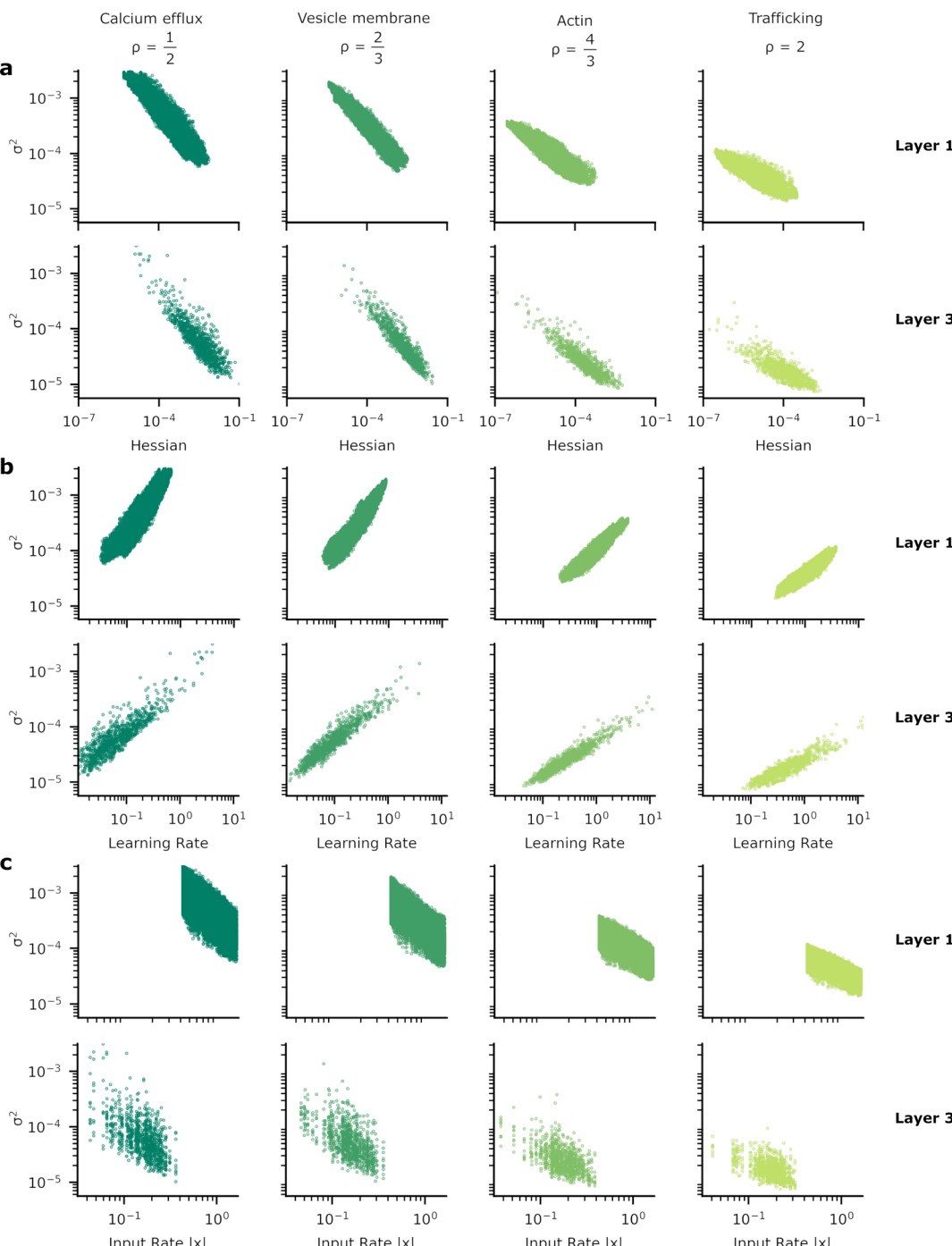

**Appendix 6—figure 1.** Patterns of synapse variability for the remaining layers of the neural network used to provide our results. In *Figure 5* we showed the heterogeneous patterns of synapse variability for the synapses connecting the two hidden layers of our artificial neural network (ANN). Here, we exhibit the equivalent plots for the other synapses, those between the input and the first hidden layer (layer 1) and from the final hidden layer to the output layer (layer 3). As in *Figure 5* we show the relationship between synaptic variance and (**a**) the Hessian; (**b**) learning rate; and (**c**) input rate. The patterns do not appear substantially different from layer to layer.

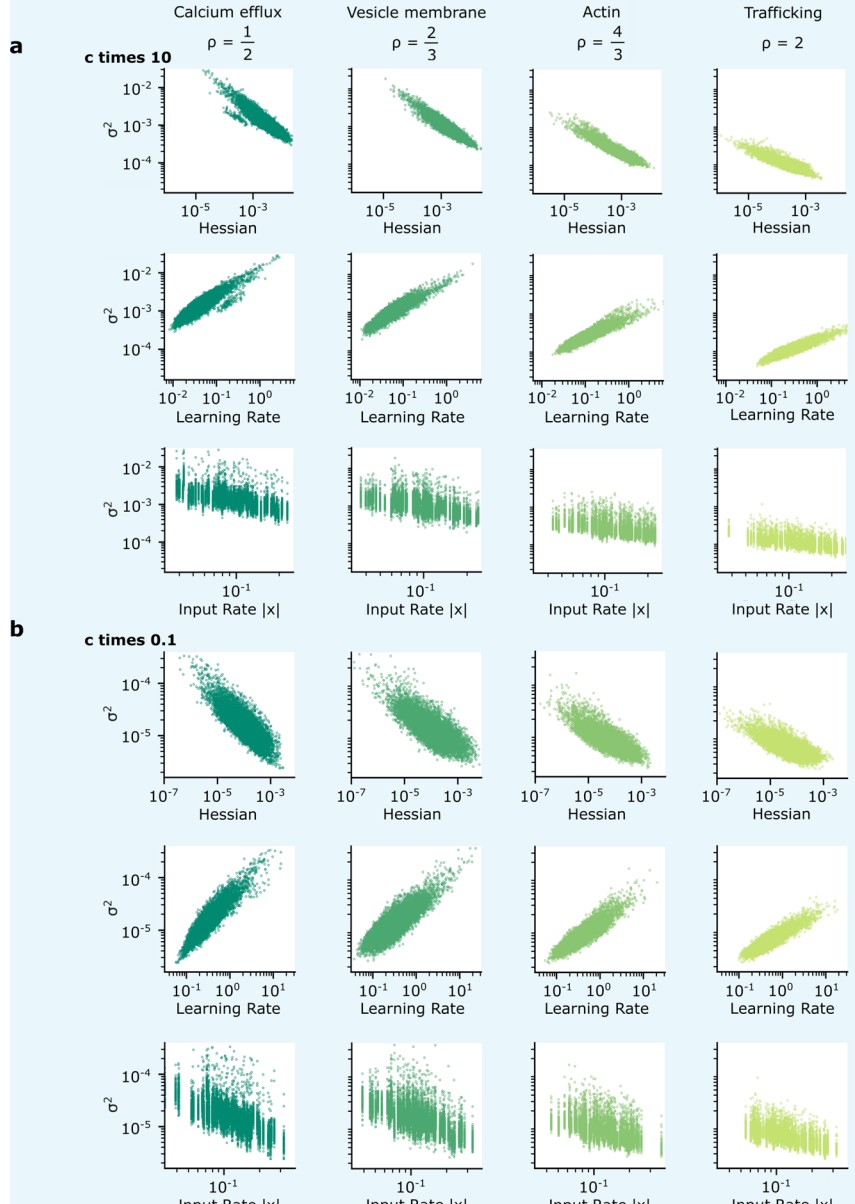

**Appendix 6—figure 2.** Patterns of synapse variability are robust to changes in the reliability. We show that the patterns of variability of synapses connecting the two hidden layers presented in *Figure 5* are preserved over a wide range of $c$. (**a, b**) When the reliability cost multiplier, $c$, is either increased (**a**) or decreased (**b**) by a factor of 10, overall synapse variability increases or decreases accordingly, but the qualitative correlations seen in *Figure 5* are preserved.

