## [Editor Report · eLife assessment]

This **important** study provides deep insight into a ubiquitous, but poorly understood, phenomenon: synaptic noise (primarily due to failures). Through a combination of theoretical analysis, simulations, and comparison to existing experimental data, this paper makes a **compelling** case that synapses are noisy because reducing noise is expensive. It touches on probably the most significant feature of living organisms -- their ability to learn -- and will be of broad interest to the neuroscience community.

---

## [Referee Report · Reviewer #1 (Public Review)]

Summary:

Given the cost of producing action potentials and transmitting them along axons, it has always seemed a bit strange that there are synaptic failures: when a spike arrives at a synapse, about half the time nothing happens. This paper proposes a perfectly reasonable explanation: reducing failures (or, more generally, reducing noise) is costly. Four possible mechanisms are proposed, each associated with a different cost, with costs of the form 1/sigma_i^rho where sigma_i is the failure-induced variability at synapse i and rho is an exponent. The four different mechanisms produce four different values of rho.

What is interesting about the study is that the model makes experimental predictions about the relationship between learning rate, variability and presynaptic firing rate. Those predictions are consistent with experimental data, making it a strong candidate model. The fact that the predictions come from reasonable biological mechanisms make it a very strong candidate model and suggest several experiments to test it further.

Interestingly, the predictions made by this model are nearly indistinguishable from the predictions made by a normative model Synaptic plasticity as Bayesian inference. Aitchison it al., Nature Neurosci. 24:565-571 (2021). As pointed out by the authors, working out whether the brain is using Bayesian inference to tune learning rules, or it just looks like it's Bayesian inference but the root cause is cost minimization, will be an interesting avenue for future research.

Finally, the authors relate their cost of reliability to the cost used in variational Bayesian inference. Intriguingly, the biophysical cost provides an upper bound on the variational cost. This is intellectually satisfying, as it answers a "why" question: why would evolution evolve to produce the kind of costs seen in the brain?

Strengths:

This paper provides a strong mix of theoretical analysis, simulations and comparison to experiments. And the extended appendices, which are very easy to read, provide additional mathematical insight.

Weaknesses:

None.

---

## [Referee Report · Reviewer #2 (Public Review)]

Summary

This manuscript argues about the similarity between two frameworks describing synaptic plasticity. In the Bayesian inference perspective, due to the noise and the limited available pre- and postsynaptic information, synapses can only have an estimate of what should be their weight. The belief about those weights is described by their mean and variance. In the energy efficient perspective, synaptic parameters (individual means and variances) are adapted such that the neural network achieves some task while penalizing large mean weights as well as small weight variances. Interestingly, the authors show both numerically and analytically the strong link between those two frameworks. In particular, both frameworks predict that (a) synaptic variances should decrease when the input firing rate increases and (b) that the learning rate should increase when the weight variances increase. Both predictions have some experimental support.

Strengths

(1) Overall, the paper is very well written and the arguments are clearly presented.

(2) The tight link between the Bayesian inference perspective and the energy efficiency perspective is elegant and well supported, both with numerical simulations as well as with analytical arguments.

(3) I also particularly appreciate the derivation of the reliability cost terms as a function of the different biophysical mechanisms (calcium efflux, vesicle membrane, actin and trafficking). Independently of the proposed mapping between the Bayesian inference perspective and the energy efficiency perspective, those reliability costs (expressed as power-law relationships) will be important for further studies on synaptic energetics.

Weaknesses

(1) As recognised by the authors, the correspondence between the entropy term in the variational inference description and the reliability cost in the energetic description is strong, but not perfect. Indeed, the entropy term scales as -log(sigma) while reliability cost scales as sigma^(-rho).

(2) Even though this is not the main point of the paper, I appreciate the effort made by the authors to look for experimental data that could in principle validate the Bayesian/energetic frameworks. A stronger validation will be an interesting avenue for future research.

---

## [Author Response]

The following is the authors’ response to the original reviews.

Weaknesses(1) The authors face a technical challenge (which they acknowledge): they use two numbers (mean and variance) to characterize synaptic variability, whereas in the brain there are three numbers (number of vesicles, release probability, and quantal size). Turning biological constraints into constraints on the variance, as is done in the paper, seems somewhat arbitrary. This by no means invalidates the results, but it means that future experimental tests of their model will be somewhat nuanced.

Agreed. There are two points to make here.

First, the mean and variance are far more experimentally accessible than *n*, *p* and *q*. The EPSP mean and variance is measured directly in paired-patch experiments, whereas getting *n*, *p* and *q* either requires far more extensive experimentation, or making strong assumptions. For instance, the data from Ko et al. (2013) gives the EPSP mean and variance, but not (directly) *n*, *p* and *q*. Thus, in some ways, predictions about means and variances are easier to test than predictions about *n*, *p* and *q*.

That said, we agree that in the absence of an extensive empirical accounting of the energetic costs at the synapse, there is inevitably some arbitrariness as we derive our energetic costs. That was why we considered four potential functional forms for the connection between the variance and energetic cost, which covered a wide range of sensible forms for this energetic cost. Our results were robust to this wide range functional forms, indicating that the patterns we describe are not specifically due to the particular functional form, but arise in many settings where there is an energetic cost for reliable synaptic transmission.

(2) The prediction that the learning rate should increase with variability relies on an optimization scheme in which the learning rate is scaled by the inverse of the magnitude of the gradients (Eq. 7). This seems like an extra assumption; the energy efficiency framework by itself does not predict that the learning rate should increase with variability. Further work will be needed to disentangle the assumption about the optimization scheme from the energy efficiency framework.

Agreed. The assumption that learning rates scale with synapse importance is separate. However, it is highly plausible as almost all modern state-of-the-art deep learning training runs use such an optimization scheme, as in practice it learns far faster than other older schemes. We have added a sentence to the main text (line 221), indicating that this is ultimately an assumption.

Major(1) The correspondence between the entropy term in the variational inference description and the reliability cost in the energetic description is a bit loose. Indeed, the entropy term scales as −log(σ) while reliability cost scales as σ−ρ. While the authors do make the point that σ−ρ upper bounds −log(σ) (up to some constant), those two cost terms are different. This raises two important questions:a. Is this difference important, i.e. are there scenarios for which the two frameworks would have different predictions due to their different cost functions?b. Alternatively, is there a way to make the two frameworks identical (e.g. by choosing a proposal distribution Q(w) different from a Gaussian distribution (and tuneable by a free parameter that could be related to *ρ*) and therefore giving rise to an entropy term consistent with the reliability cost of the energy efficiency framework)?

To answer b first, there is no natural way to make the two frameworks identical (unless we assume the reliability cost is proportional to log_σsyn_, and we don’t think there’s a biophysical mechanism that would give rise to such a cost). Now, to answer a, in Fig. 7 we extensively assessed the differences between the energy efficient *σsyn* and the Bayesian *σpost*. In Fig.7bc, we find that *σsyn* and *σpost* are positively correlated in all models. This positive correlation indicates that the qualitative predictions made by the two frameworks (Bayesian inference and energy efficiency) are likely to be very similar. Importantly though, there are systematic differences highlighted by Fig. 7ab. Specifically, the energy efficient *σsyn* tends to vary less than the Bayesian *σpost*. This appears in Fig. 7b which shows the relationship between *σsyn* (on the y-axis) and *σpost* (on the x-axis). Specifically, this plot has a slope that is smaller than one for all our models of the biophysical cost. Further, the pattern also appears in the covariance ellipses in Fig. 7a, in that the Bayesian covariance ellipses tend to be long and thin, while the energy efficient covariance ellipsis are rounder. Critically though both covariance ellipses show the same pattern in that there is more noise along less important directions (as measured by the Hessian).

We have added a sentence (line 273) noting that the search for a theoretical link is motivated by our observations in Fig. 7 of a strong, but not perfect link between the pattern of variability predicted by Bayesian and energy-efficient synapses.

(2) Even though I appreciate the effort of the authors to look for experimental evidence, I still find that the experimental support (displayed in Fig. 6) is moderate for three reasons.a. First, the experimental and simulation results are not displayed in a consistent way. Indeed, Fig 6a displays the relative weight change |*Dw*|*/w* as a function of the normalised variability *σ2/*|*µ*| in experiments whereas the simulation results in Fig 5c display the variance *σ2 as a function of the learning rate. Also, Fig 6b displays the normalised variability σ2/|µ*| as a function of the input rate whereas Fig 5b displays the variance _σ_2 as a function of the input rate. As a consequence the comparison between experimental and simulation results is difficult.b. Secondly, the actual power-law exponents in the experiments (see Fig 6a resp. 6b) should be compared to the power-law exponents obtained in simulation (see Fig 5c resp. Fig 5b). The difficulty relies here on the fact that the power-law exponents obtained in the simulations directly depend on the (free) parameter *ρ*. So far the authors precisely avoided committing to a specific *ρ*, but rather argued that different biophysical mechanisms lead to different reliability exponents *ρ*. Therefore, since there are many possible exponents *ρ* (and consequently many possible power-law exponents in simulation results in Fig 5), it is likely that one of them will match the experimental data. For the argument to be stronger, one would need to argue which synaptic mechanism is dominating and therefore come up with a single prediction that can be falsified experimentally (see also point 4 below).c, Finally, the experimental data presented in Fig6 are still “clouds of points". A coefficient of *r* = 0_.*52 (in Fig 6a) is moderate evidence while the coefficient of r = −0*._26 (in Fig 6b) is weak evidence.

The key thing to remember is that our paper is not about whether synapses are “really" Bayesian or energy efficient (or both/neither). Instead, the key point of our paper, as expressed in the title, is to show that the experimental predictions of Bayesian synapses are very similar to the predictions from energy efficient synapses. And therefore energy efficient synapses are very difficult to distinguish experimentally from Bayesian synapses. In that context, the two plots in Fig. 6 are not really intended to present evidence in favour of the energy efficiency / Bayesian synapses. In fact, Fig. 6 isn’t meant to constitute a contribution of the paper at all, instead, Fig. 6 serves merely as illustrations of the kinds of experimental result that have (Aitchison et al. 2021) or might (Schug et al. 2021) be used to support Bayesian synapses. As such, Fig. 6 serves merely as a jumping-off point for discussing how very similar results might equally arise out of Bayesian and energy-efficiency viewpoints.

We have modified our description of Fig. 6 to further re-emphasise that the panels in Fig. 6 is not our contribution, but is taken directly from Schug et al. 2021 and Aitchison et al. 2021 (we have also modified Fig 6 to be precisely what was plotted in Schug et al. 2021, again to re-emphasise this point). Further, we have modified the presentation to emphasise that these plots serve merely as jumping off points to discuss the kinds of predictions that we might consider for Bayesian and energy efficient synapses.

This is important, because we would argue that the “strength of support" should be assessed for our key claim, made in the title, that “Signatures of Bayesian inference emerge from energy efficient synapses".

a) To emphasise that these are previously published results, we have chosen axes to matchthose used in the original work (Aitchison et al. 2021) and (Schug et al. 2021).

b) We agree that a close match between power-law exponents would constitute strong evidencefor energy-efficiency / Bayesian inference, and might even allow us to distinguish them. We did consider such a comparison, but found it was difficult for two reasons. First, while the confidence intervals on the slopes exclude zero, they are pretty broad. Secondly, while the slopes in a one-layer network are consistent and match theory (Appendix 5) the slopes in deeper networks are far more inconsistent. This is likely to be due to a number of factors such as details of the optimization algorithm and initialization. Critically, if details of the optimization algorithm matter in simulation, they may also matter in the brain. Therefore, it is not clear to us that a comparison of the actual slopes is can be relied upon.

To reiterate, the point of our article is not to make judgements about the strength ofevidence in previously published work, but to argue that Bayesian and energy efficient synapses are difficult to distinguish experimentally as they produce similar predictions. That said, it is very difficult to make blanket statements about the strength of evidence for an effect based merely on a correlation coefficient. It is perfectly possible to have moderate correlation coefficients along with very strong evidence of an effect (and e.g. very strong p-values), e.g. if there is a lot of data. Likewise, it is possible to have a very large correlation coefficient along with weak evidence of an effect (e.g. if we only have three or four datapoints, which happen to lie in a straight line). A small correlation coefficient is much more closely related to the effect-size. Specifically, the effect-size, relative to the “noise", which usually arises from unmeasured factors of variation. Here, we know there are many, many unmeasured factors of variation, so even in the case that synapses are really Bayesian / energy-efficient, the best we can hope for is low correlation coefficients

As mentioned in the public review, a weakness in the paper is the derivation of the constraints on σi given the biophysical costs, for two reasons.a.First, it seemed a bit arbitrary whether you hold n fixed or p fixed.b.Second, at central synapses, n is usually small – possibly even usually 1: REF(Synaptic vesicles transiently dock to refill release sites, Nature Neuroscience 23:1329-1338, 2020); REF(The ubiquitous nature of multivesicular release Trends Neurosci. 38:428-438, 2015). Fixing n would radically change your cost function. Possibly you can get around this because when two neurons are connected there are multiple contacts (and so, effectively, reasonably large n). It seems like this is worth discussing.

a) Ultimately, we believe that the “real” biological cost function is very complex, and most likely cannot be written down in a simple functional form. Further, we certainly do not have the experimental evidence now, and are unlikely to have experimental evidence for a considerable period into the future to pin down this cost function precisely. In that context, we are forced to resort to two strategies. First, using simplifying assumptions to derive a functional form for the cost (such as holding *n* or *p* fixed). Second, considering a wide range of functional forms for the cost, and ensuring our argument works for all of them.

b) We appreciate the suggestion that the number of connections could be used as a surrogate where synapses have only a single release site. As you suggest we can propose an alternative model for this case where *n* represents the number of connections between neurons. We have added this alternative interpretation to our introduction of the quantal model under title “Biophysical costs". For a fixed PSP mean we could either have many connections with small vesicles or less connections with larger vesicles. Similarly for the actin cost we would certainly require more actin if the number of connections were increased.

Minor(1) A few additional references could further strengthen some claims of the paper:Davis, Graeme W., and Martin Muller. “Homeostatic Control of Presynaptic Neurotransmitter Release." Annual Review of Physiology 77, no. 1 (February 10, 2015): 251-70. https://doi.org/10.1146/annurev-physiol-021014-071740. This paper provides elegant experimental support for the claim (in line 538 now 583) that *µ* is kept constant and q acts as a compensatory variable.Jegminat, Jannes, Simone Carlo Surace, and Jean-Pascal Pfister. “Learning as Filtering: Implications for Spike-Based Plasticity." Edited by Blake A Richards. PLOS Computational Biology 18, no. 2 (February 23, 2022): e1009721. https://doi.org/10.1371/journal.pcbi.1009721.This paper also showed that a lower uncertainty implies a lower learning rate (see e.g. in line 232), but in the context of spiking neurons.

Figure 1 of the the first suggested paper indeed shows that quantal size is a candidate for homeostatic scaling (fixing *µ*). This review also references lots of further evidence of quantal scaling and evidence for both presynaptic and postsynaptic scaling of *q* leaving space for speculation on whether vesicle radius or postsynaptic receptor number is the source of a compensatory *q*. On line 583 we have added a few lines pointing to the suggested review paper.

The second reference demonstrates Bayesian plasticity in the context of STDP, proposing learning rates tuned to the covariance in spike timing. We have added this as extra support for assuming an optimisation scheme that tunes learning rates to synapse importance and synapse variability (line 232).

In the numerical simulations, the reliability cost is implemented with a single power-law expression (reliabilitycost=cσ−μ). However, in principle, all the reliability costs will play in conjunction, i.e. reliability cost =∑iciσ−ρi. While I do recognise that it may be difficult to estimate the biophysical values of the various *ci*, it might be still relevant to comment on this.

Agreed. Limitations in the literature meant that we could only form a cursory review of the relative scale of each cost using estimates by Atwell, (2001), Engl, (2015). On line 135 we have added a paragraph explaining the rationale for considering each cost independently.

(3) In Eq. 8: σ^2^ doesn’t depend on variability in *q*, which would add another term; barring algebra mistakes, it’s σ2=n2p2var⁡[q]+np(1−p)<q2>=np(1+(n−1)p)var⁡[q]+np(1−p)<q>2. It seems worth mentioning why you didn’t include it. Can you argue that it’s a small effect?

Agreed. Ultimately, we dropped this term because we expected it to be small relative to variability in vesicle release, and because it would be difficult to quantify In practice, the variability is believed to be contributed mostly by variability in vesicle release. The primary evidence for this is histograms of EPSP amplitudes which show classic multi-peak structure, corresponding to one, two three etc. EPSPs. Examples of these plots include:

- “The end-plate potential in mammalian muscle”, Boyd and Martin (1956); Fig. 8.

- “Structure and function of a neocortical synapse”, Holler-Rickauer et al. (2019); Extended Figure 5.

(3) On pg. 7 now pg. 8, when the Hessian is introduced, why not say what it is? Or at least the diagonal elements, for which you just sum up the squared activity. That will make it much less mysterious. Or are we relying too much on the linear model given in App 2? If so, you should tell us how the Hessian was calculated in general. Probably in an appendix.

With the intention of maintaining the interest of a wide audience we made the decision to avoid a mathematical definition of the Hessian, opting instead for a written definition i.e. line 192 - “*Hii*; the second derivatives of the objective with respect to *wi*.” and later on a schematic (Fig. 4) for how the second derivative can be understood as a measure of curvature and synapse importance. Nonetheless, this review point has made us aware that the estimated Hessian values plotted in Fig. 5a have been insufficiently explained so we have added a reference on line 197 to the appendix section where we show how we estimated the diagonal values of the Hessian.

(4) Fig. 5: assuming we understand things correctly, Hessian ∝ |*x*|2. Why also plot _σ_2 versus |*x*|? Or are we getting the Hessian wrong?

The Hessian is proportional to ∑t|xt|2. If you assume that time steps are small and neurons spike, then xt∈{0,1}, and |xt|2=|xt|. it is difficult to say what timestep is relevant in practice.

(5) To get Fig. 6a, did you start with Fig. Appendix 1-figure 4 from Schug et al, and then use σ2/μ=(1−p)q, drop the *q*, and put 1 − *p* on the x-axis? Either way, you should provide details about where this came from. It could be in Methods.

We have modified Fig. 6 to use the same axes as in the original papers.

(6) Lines 190-3: “The relationship between input firing rate and synaptic variability was first observed by Aitchison et al. (2021) using data from Ko et al. (2013) (Fig. 6a). The relationship between learning rate and synaptic variability was first observed by Schug et al. (2021), using data from Sjostrom et al. (2003) as processed by Costa et al. (2017) (Fig. 6b)." We believer 6a and 6b should be interchanged in that sentence.

Thank you. We have switched the text appropriately.

(7) What is posterior variance? This seems kind of important.

This refers to the “posterior variance" obtained using a Bayesian interpretation of the problem of obtaining good synaptic weights (Aitchison et al. 2021). In our particular setting, we estimate posterior variances by setting up the problem as variational inference: see Appendix 4 and 5, which is now referred to in line 390.

(8) Lines 244-5: “we derived the relationships between the optimized noise, *σi* and the posterior variable, *σpost* as a function of *ρ* (Fig. 7b;) and as a function of c (Fig. 7c)." You should tell the reader where you derived this. Which is Eq. 68c now 54c. Except you didn’t actually derive it; you just wrote it down. And since we don’t know what posterior variance is, we couldn’t figure it out.

If *H* is the Hessian of the log-likelihood, and if the prior is negligable relative to the the likelihood, then we get Eq. 69c. We have added a note on this point to the text.

(9) We believe Fig. 7a shows an example pair of synapses. Is this typical? And what about Figs. 7b and c. Also an example pair? Or averages? It would be helpful to make all this clear to the reader.

Fig. 7a shows an illustrative pair of synapses, chosen to best display the relative patterns of variability under energy efficient and Bayesian synapses. We have noted this point in the legend for Fig. 7. Fig. 7bc show analytic relationships between energy efficient and Bayesian synapses, so each line shows a whole continuum of synapses(we have deleted the misleading points at the ends of the lines in Fig. 7bc).

(10) The y-axis of Fig 6a refers to the synaptic weight as w while the x-axis refers to the mean synaptic weight as mu. Shouldn’t it be harmonised? It would be particularly nice if both were divided by *µ*, because then the link to Fig. 5c would be more clear.

We have changed the y-axis label of Fig. 6a from *w* to *µ*. Regarding the normalised variance, we did try this but our Gaussian posteriors allowed the mean to become small in our simulations, giving a very high normalised variance. To remedy this we would likely need to assume a log- posterior, but this was out of scope for the present work.

(11) Line 250 (now line 281): “Finally, in the Appendix". Please tell us which Appendix. Also, why not point out here that the bound is tightest at small *ρ*?

We have added the reference to the the section of the appendix with the derivation of the biological cost as a bound on the ELBO. We have also referenced the equation that gives the limit of the biological cost as ρ tends to zero.

(12) When symbols appear that previously appeared more than about two paragraphs ago, please tell us where they came from. For instance, we spent a lot of time hunting for *ηi*. And below we’ll complain about undefined symbols. Which might mean we just missed them; if you told us where they were, that problem would be eliminated.

We have added extra references for the symbols in the text following Eq. 69.

(13) Line 564, typo (we think): should be σ−2.

Good spot. This has been fixed.

(14) A bit out of order, but we don’t think you ever say explicitly that *r* is the radius of a vesicle. You do indicate it in Fig. 1, but you should say it in the main text as well.

We have added a note on this to the legend in Fig. 1.

(15) Eq. 14: presumably there’s a cost only if the vesicle is outside the synapse? Probably worth saying, since it’s not clear from the mechanism.

Looking at Pulido and Ryan (2021) carefully, it is clear that they are referring to a cost for vesicles inside the presynaptic side of the synapse. (Importantly, vesciles don’t really exist outside the synapse; during the release process, the vesicle membrane becomes part of the cell membrane, and the contents of the vesicle is ejected into the synaptic cleft).

(16) App. 2: why solve for mu, and why compute the trace of the Hessian? Not that it hurts, but things are sort of complicated, and the fewer side points the better.

Agreed, we have removed the solution for μ, and the trace, and generally rewritten Appendix 2 to clarify definitions, the Hessian etc.

(17) Eq. 35: we believe you need a minus sign on one side of the equation. And we don’t believe you defined p(d|w). Also, are you assuming g = partial log p(d|w)/partial w? This should be stated, along with its implications. And presumably, it’s not really true; people just postulate that *p*(*d*|*w*) ∝ exp(−log_loss_)?

We have replaced p(d|w) with p(y, x|w), and we replaced “overall cost” with log P(y|w, x). Yes, we are also postulating that p(y|w, x) ∝ exp(−log loss), though in our case that does make sense as it corresonds to a squared loss.

As regards the minus sign, in the orignal manuscript, we had the second derivative of the cost. There is no minus sign for the cost, as the Hessian of the cost at the mode is positive semi-definite. However, once we write the expression in terms of a log-likelihood, we do need a minus sign (as the Hessian of the log-likelihood at a mode is negative semi-definite).

(18) Eq. 47 now Eq. 44: first mention of *CBi*;*i*?

We have added a note describing CB around these equations.

(19) The “where" doesn’t make sense for Eqs. 49 and 50; those are new definitions.

We have modified the introduction of these equations to avoid the problematic “where”.

(20) Eq. 57 and 58 are really one equation. More importantly: where does Eq. 58 come from? Is this the *H* that was defined previously? Either way, you should make that clear.

We have removed the problematic additional equation line number, and added a reference to where H comes from.

(21) In Eq. 59 now Eq. 60 aren’t you taking the trace of a scalar? Seems like you could skip this.

We have deleted this derivation, as it repeats material from the new Appendix 2.

(22) Eq. 66 is exactly the same as Eq. 32. Which is a bit disconcerting. Are they different derivations of the same quantity? You should comment on this.

We have deleted lots of the stuff in Appendix 5 as, we agree, it repeats material from Appendix 2 (which has been rewritten and considerably clarified).

(23) Eq. 68 now 54, left column: please derive. we got:gai = gradient for weight i on trial a=(va−waxai)xai/ϵ2where the second equality came from Eq. 20. Thus<gai2>=<(va−waxai)2xai2>/ϵ4≈<(va−waxai)2><xai2>/ϵ4=<xai2>/ϵ2∝HiiIs that correct? If so, it’s a lot to expect of the reader. Either way, a derivation wouldbe helpful.

We agree it was unnecessary and overly complex, so we have deleted it.

(24) App 5–Figure 2: presumably the data for panel b came from Fig. 6a, with the learning rate set to Δw/w? And the data for panel c from Fig. 6b? This (or the correct statement, if this is wrong) should be mentioned.

Yes, the data for panel c came from Fig. 6b. We have deleted the data in panel b, as there are some subtleties in interpretation of the learning rates in these settings.

(25) line 952 now 946: typo, “and the from".

Corrected to “and from".